# Moshi$_{\text{A}}^{\text{R}}$: Asynchronous Knowledge Retrieval for Full-Duplex Speech Language Models

**Chung-Ming Chien** [1 2 *]  **Manu Orsini** [2]  **Eugene Kharitonov** [2 3 **]  **Neil Zeghidour** [2 3]  **Karen Livescu** [1]
**Alexandre Défossez** [2 3]

## Abstract

Speech-to-speech language models have recently emerged to enhance the naturalness of conversational AI. In particular, full-duplex models are distinguished by their real-time interactivity, including handling of pauses, interruptions, and backchannels. However, improving their factuality remains an open challenge. While scaling the model size could address this gap, it would make real-time inference prohibitively expensive. In this work, we propose Moshi$_{\text{A}}^{\text{R}}$, a modular approach that combines a compact full-duplex interface with selective retrieval to access more powerful knowledge sources. Our asynchronous framework enables the model to identify knowledge-demanding queries and ground its responses in external information. By leveraging the natural temporal gap between response onset and the delivery of core information, the retrieval process can be completed while maintaining a natural conversation flow. With this approach, Moshi$_{\text{A}}^{\text{R}}$ achieves factuality comparable to the best publicly released non-duplex speech language models while preserving the interactivity inherent to full-duplex systems. Moreover, our flexible design supports plug-and-play retrieval methods without retraining and demonstrates strong performance on out-of-domain mathematical reasoning tasks.

## 1. Introduction

Building voice interfaces for artificial intelligence (AI) systems capable of assisting humans across a wide range of sce-

narios has long been central to visions of future technology. A user-friendly voice interface should create a natural conversation experience, allowing users to communicate with AI systems as if they were speaking to a real human assistant. Earlier approaches typically combined multiple components – such as automatic speech recognition (ASR), text-based dialogue management, and text-to-speech (TTS) synthesis – and optimized them for conversational use cases (Seneff et al., 1998; Levin et al., 2000; Bohus & Rudnicky, 2009). More recent research has shifted toward end-to-end approaches to avoid information loss introduced by speech-to-text conversion, such as prosody, rhythm, and intonation, while also reducing latency and friction caused by cascaded pipelines (Zhang et al., 2023; Nachmani et al., 2024; Xie & Wu, 2024; Fang et al., 2025a; Zeng et al., 2024).

Among modern frameworks, **full-duplex** models (Défossez et al., 2024; Yu et al., 2025) are distinguished by their ability to "listen while speaking," in contrast to turn-based methods that process speech in large chunks (e.g., sentences) and allow transitioning between listening and speaking states only after each chunk is completed (see Figure 1). The capability to concurrently receive speech inputs and generate responses enables full-duplex models to react more promptly to user inputs (Zhang et al., 2025; Chen et al., 2025a) and can better model the complex **interactivity** of real-world conversation (Veluri et al., 2024; Yu et al., 2025; Roy et al., 2026). However, the full-duplex approach also introduces unique challenges such as the need for real-time speech processing and generation. Meanwhile, recent studies indicate that native audio models struggle more than text models with tasks requiring **factuality**, such as question answering (Wang et al., 2025a). This reduced factuality is at least in part due to the much smaller amounts of speech data than text data (in terms of number of words) available for training.

To address the challenge of improving factuality while maintaining interactivity, we propose Moshi$_{\text{A}}^{\text{R}}$, the first full-duplex voice model equipped with retrieval-augmented generation (RAG) capability, built as an extension of the full-duplex speech LM Moshi (Défossez et al., 2024). While RAG has become a widely adopted technique for enhancing the factuality of large language models (LLMs) (Lewis

---

*Work done when visiting Kyutai **Work done while at Kyutai [1]Toyota Technological Institute at Chicago, Chicago, USA [2]Kyutai, Paris, France [3]Gradium, Paris, France. Correspondence to: Chung-Ming Chien <cmchien@ttic.edu>, Alexandre Défossez <alex@kyutai.org>.

*Proceedings of the 43$^{rd}$ International Conference on Machine Learning*, Seoul, South Korea. PMLR 306, 2026. Copyright 2026 by the author(s).

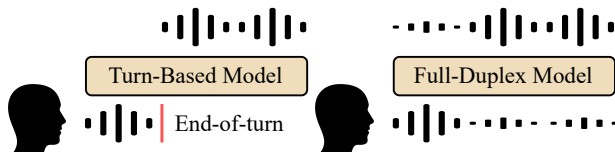

*Figure 1.* Illustration of turn-based models versus full-duplex models. The former must explicitly switch between speaking and listening states, while the latter can concurrently speak and listen.

et al., 2020), its integration into full-duplex voice systems remains largely unexplored due to the strict real-time constraints imposed by continuous speech interaction. We tackle this challenge by exploiting the natural temporal gap between the onset of a spoken response and the emergence of its key informational content (the "keyword delay" in Figure 2). Leveraging this observation, we design specialized fine-tuning data that trains Moshi to predict a retrieval trigger signal when the user poses knowledge-intensive queries. This signal asynchronously invokes an information retrieval system to generate reference documents relevant to the conversation context. The retrieved information is then incorporated into the response generation process before the key content is reached. We design the RAG mechanism so as to guarantee that the entire retrieval process completes within two seconds – shorter than the keyword delay of many existing speech LMs (see Table 1). In addition to improving factuality without compromising interactivity, $\text{Moshi}_{AG}^{R}$ is retrieval-back-end agnostic, enabling seamless integration of different retrieval methods – such as LLM-based retrievers or search engines – as long as they can provide textual references within a reasonable time. This design offers flexibility and extensibility for future improvements.

Experimental results demonstrate that $\text{Moshi}_{AG}^{R}$ significantly improves the factuality of Moshi on question answering (QA) benchmarks while maintaining good interactivity in speech conversation as measured by full-duplex benchmarks (Lin et al., 2025; 2026). We further show that performance can be enhanced at inference time by simply switching to more powerful retrieval back ends without retraining the base model. Finally, we demonstrate that $\text{Moshi}_{AG}^{R}$ generalizes well to previously unseen mathematical reasoning tasks, which are challenging for both the original Moshi and other speech LMs. This can be viewed as an early exploration of the tool-use capabilities of full-duplex models, where Moshi effectively leverages an LLM as an external tool to solve mathematical tasks. Our results suggest the broader potential for enabling general tool use in full-duplex models and demonstrate the promise of building more powerful, reliable, and user-friendly voice AI assistants by combining real-time interactive voice interfaces with more capable problem-solving mechanisms.

We release $\text{Moshi}_{AG}^{R}$'s inference code[1] along with several demo videos[2] for public access.

## 2. Related Work

Since dGSLM (Nguyen et al., 2023) initiated research on end-to-end multi-speaker conversational modeling (Veluri et al., 2024; Wang et al., 2025b), duplex models have emerged as an increasingly prominent direction. To jointly model user and system speech, one line of work adopts time-multiplexing approaches (Zhang et al., 2025; Chen et al., 2025a; Mai & Carson-Berndsen, 2025), in which the model alternates between processing fixed-duration chunks of user input and generating responses of the same duration. In contrast, models with a dual-channel architecture like Moshi (Défossez et al., 2024; Yu et al., 2025; Hu et al., 2025; Yao et al., 2025; Roy et al., 2026) enable high frame-rate, simultaneous modeling of input and output speech streams.

To improve the factuality of speech dialogue models, recent works have incorporated RAG (Min et al., 2025; Rackauckas & Hirschberg, 2025; Chen et al., 2025b; Feng et al., 2025). Concurrent works such as StreamRAG (Arora et al., 2025) and KAME (Kuroki et al., 2026) also exploit temporal gaps in spoken conversations to perform information retrieval. However, StreamRAG is restricted to non-full-duplex settings with fixed, pre-indexed corpora, failing to address the strict timing constraints of real-time full-duplex interaction. While KAME supports full-duplex modeling, it is limited to LLM-based information sources and relies on frequent, fixed-interval LLM calls that are agnostic to conversational context to generate content, leading to significant computational waste. In contrast, $\text{Moshi}_{AG}^{R}$ provides a more efficient alternative by dynamically triggering retrieval only as needed. By supporting both LLM-based knowledge and direct web-search capabilities, $\text{Moshi}_{AG}^{R}$ extends the RAG paradigm to open-domain full-duplex conversation. Beyond RAG, alternative approaches such as chain-of-thought reasoning for audio and speech models (Zhifei et al., 2025; Ma et al., 2025; Chiang et al., 2026; 2025; Shih et al., 2026) have also been explored; these techniques are complementary to our framework and could be naturally combined in future work.

## 3. System Design

The $\text{Moshi}_{AG}^{R}$ framework is built upon Moshi (Défossez et al., 2024). To integrate external information into Moshi's response generation, we first analyze the timing constraints in human-machine speech conversations. Based on it, we propose a framework consisting of a full-duplex front end and an asynchronous retrieval back end that operate in

---

[1]https://github.com/kyutai-labs/moshi-rag
[2]https://kyutai.org/blog/2026-04-30-moshi-rag

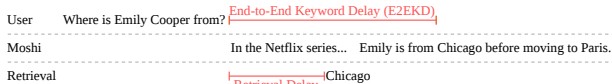

| User | Where is Emily Cooper from? | End-to-End Keyword Delay (E2EKD) | |
|---|---|---|---|
| Moshi | | In the Netflix series... Emily is from Chicago before moving to Paris. |
| Retrieval | | Retrieval Delay | Chicago |

*Figure 2.* Different types of delays in human-machine conversations. End-to-end keyword delay (E2EKD) measures the time between the end of the user's question and the most informative word in the response. Retrieval delay measures how long it takes for the back end to provide relevant information.

parallel, enabling the model to maintain interactivity while incorporating externally retrieved knowledge in real time.

### 3.1. Timing Constraints

Below, we introduce some terminology related to latency in human-machine conversation (illustrated in Figure 2):

- **Time-to-first-audio-token (TTFAT)**: the audio-domain counterpart of the commonly used time-to-first-token (TTFT) metric for LLMs. We define TTFAT as the delay between the end of a user's utterance and the moment the model generates the first audio token of its response.[3]
- **Keyword delay:** time interval from the beginning of the model's spoken response to the point at which the key content (i.e., a keyword that directly answer the user's query, if any) first appears. See Section 5.2 for details.
- **End-to-end keyword delay (E2EKD):** the total time from the end of the user's query to the moment the keyword is mentioned in the model's response. By definition, E2EKD is the sum of TTFAT and keyword delay.
- **Retrieval delay:** the time from the prediction of a retrieval trigger to the completion of the retrieval process.

E2EKD is a critical perceptual metric, as it determines how quickly meaningful information is delivered to the user. For retrieval-augmented systems, assuming that the retrieval is not triggered before the user query finishes, the retrieval delay must be shorter than the E2EKD in order for the retrieved information to be integrated into the response in time. Our preliminary analysis shows that the E2EKD of existing speech LMs often exceeds 3 seconds (see Table 1). Accordingly, we target for Moshi$_{AG}^{R}$ a retrieval delay of no more than 2 seconds during both data construction and model training, ensuring that external knowledge can be effectively integrated without compromising real-time interaction quality.

### 3.2. System Overview

In this paper, we define the **front end** as the modules that directly receive or generate audio to communicate with the user in real time, while the **back end** consists of components that do not directly interact with the user. For

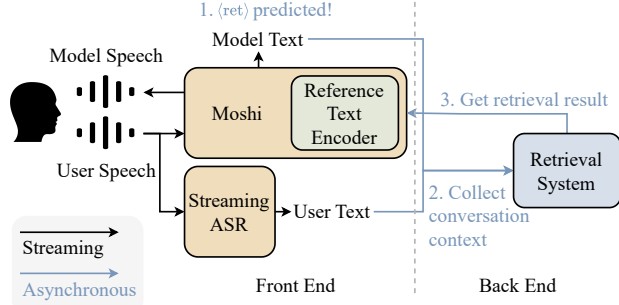

*Figure 3.* Illustration of the front-end and back-end components in Moshi$_{AG}^{R}$. When the model needs external information, it outputs a ⟨ret⟩ token. The conversation transcript is sent to the back end which operates asynchronously. Once ready, the result is injected into Moshi which then adapts its response with no interruption.

example, in a traditional cascaded ASR–dialogue–TTS system, the ASR and TTS modules are front-end components by this definition, whereas the text-based dialogue management system belongs to the back end. To optimize user experience, the front end must provide immediate feedback and reactions to user inputs. In contrast, the back end can prioritize factuality and reasoning, such as planning dialogue flow, selecting correct information, or managing topics, and benefits from greater time flexibility since it does not operate under strict real-time constraints.

In this work, we use the original Moshi model (with minor modifications) as the full-duplex front end, while an asynchronous information retrieval system operates in parallel as the back end. Additionally, since most information retrieval systems are text-based, an additional streaming ASR model is used to transcribe user speech into text for retrieval purposes.[4] This ASR model directly receives speech inputs and thus, by definition, is part of the front end. Figure 3 provides a conceptual overview of the system. The lack of synchronization between the front end and back end allows the system to effectively "think while listening and speaking," similar to human's cognitive abilities.

During a speech conversation, the front-end Moshi takes user speech tokens encoded by the Mimi codec encoder (Défossez et al., 2024) as input and autoregressively predicts both textual transcriptions (with padding tokens inserted) and corresponding speech tokens for the model's response in separate channels. The only modifications from the original Moshi model are the introduction of a special retrieval trigger token ⟨ret⟩ and a reference text encoder. As shown in Figures 3 and 4, when the ⟨ret⟩ token is predicted, we collect the textual transcriptions of both the user and the assistant from the ASR and Moshi outputs,

---

[3]This definition focuses on content generation latency and excludes the time for token-to-waveform conversion, e.g. the codec or vocoder, which is orthogonal to the scope of this work.

[4]Although it is possible to build the transcription functionality into the main Moshi model, we use a separate ASR model to minimize training efforts.

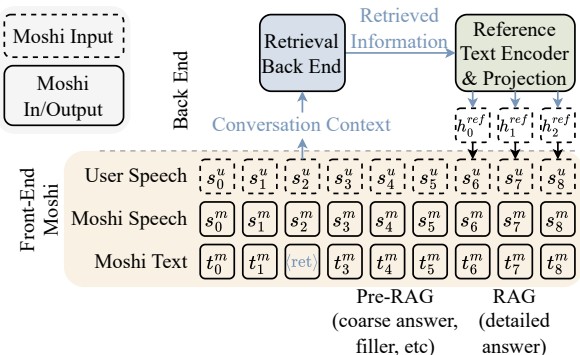

*Figure 4.* Text and audio token streams of the inputs and outputs of Moshi$_{AG}^{R}$. Front-end Moshi receives at all time its previous step token predictions and the user speech tokens. When the retrieval result is ready, its representation is summed with the embeddings from other token streams and ingested over a number of time steps.

respectively, and pass the aggregated conversational context to the retrieval back end. While retrieval is in progress, the front-end Moshi continues to operate in full duplex, receiving incoming speech and generating responses so that the conversation proceeds without interruption.

We refer to the content generated after $\langle\text{ret}\rangle$ is predicted but before the retrieval process completes as **pre-RAG content**. In our training data, pre-RAG content typically includes a coarse answer to the user's query as well as conversational filler phrases (e.g., "Let me check that for you...") that do not require domain-specific knowledge to generate. Once the retrieval process completes, the retrieved document is encoded with a reference text encoder and then injected into Moshi. This allows the model to ground the remaining parts of its response in external knowledge and thus provide a more accurate and detailed answer following pre-RAG content.

## 3.3. Building Blocks

As shown in Figure 3, Moshi$_{AG}^{R}$ consists of three main components: a 7B Moshi model fine-tuned with RAG training data, a 1B streaming ASR model, and a retrieval back end. Communication between these components is conducted entirely in text format. This modular design enables independent training of each component and provides flexibility to upgrade any part of the system without affecting the others. Details about individual components are described below.

### 3.3.1. RAG AUGMENTED MOSHI MODEL

The original Moshi (Défossez et al., 2024) models autoregressively text and audio tokens corresponding to its speech output with an RQ-Transformer (Lee et al., 2022), consisting of a main "temporal" Transformer (Vaswani et al., 2017) operating at 12.5 Hz and a "depth" Transformer predicting 8 audio tokens per time step. The temporal Trans-

former is also input with user audio tokens. Formally, at time step $i$, the input vector $h_i$ to the temporal Transformer is:

$$h_i = \text{emb}_{\text{text},i}^{\text{model}} + \text{emb}_{\text{speech},i}^{\text{model}} + \text{emb}_{\text{speech},i}^{\text{user}} \quad (1)$$

where $\text{emb}_{m,i}^{r}$ denotes the embedding for role $r$ (model or user) and modality $m$ (text or speech) at time step $i$.[5]

When retrieval is not activated, Moshi$_{AG}^{R}$ operates like the original Moshi. When the $\langle\text{ret}\rangle$ signal is predicted at time step $i_{\text{ret}}$, assuming that the retrieval delay is $d$ seconds, the retrieved reference text is encoded as a sequence of embeddings $\text{emb}_{1:l}^{\text{ref}}$, where $l$ is the sequence length. These embeddings are projected via a one-layer trainable linear layer and summed to the temporal Transformer input in a streaming fashion over $l$ temporal steps. The resulting reference-aware input $h_i'$ is:

$$h_i' = \begin{cases} h_i + h_{i-(i_{\text{ret}}+\frac{d}{f_r})}^{\text{ref}} & \text{if } i_{\text{ret}} + \frac{d}{f_r} < i \le i_{\text{ret}} + \frac{d}{f_r} + l \\ h_i & \text{otherwise} \end{cases}$$

$$(2)$$

where $h_i^{\text{ref}} = \text{proj}(\text{emb}_i^{\text{ref}})$ and $f_r$ is Moshi's frame rate.

A potential issue arises if long reference documents are retrieved. For example, with Moshi's 12.5 Hz frame rate, a 250-token reference could correspond to a 20-second embedding sequence, far exceeding the duration of a turn in a normal speech conversation. To address this, we adopt a pre-trained sequence compression network, ARC-Encoder (Pilchen et al., 2025), to reduce the reference sequence length by a factor of four. More choices of reference encoders are further explored in Appendix B.1.

### 3.3.2. STREAMING ASR

We employ a pre-trained streaming ASR model with 0.5-second latency (Zeghidour et al., 2025) to transcribe user speech into text for retrieval purposes. The model has only 1B parameters, making its computational cost minimal relative to other components of Moshi$_{AG}^{R}$.

### 3.3.3. RETRIEVAL BACK END

Once $\langle\text{ret}\rangle$ is predicted, we first wait 0.5 seconds for the ASR model to produce a complete transcript of the user's utterance. The collected conversation transcript is then sent to the retrieval back end, which is a text-in-text-out system capable of returning reference documents within manageable time to facilitate Moshi's next response. In this work, we consider two types of retrieval back ends:

---

[5]Specifically, the text embedding is $\text{emb}_{\text{text},i}^{\text{model}} = \text{Emb}_{\text{text}}(t_i^{\text{model}})$, where $t_i^{\text{model}}$ is the model text token at step $i$ and $\text{Emb}_{\text{text}}$ is the text embedding table. The speech embedding is given by $\text{emb}_{\text{speech},i}^{r} = \text{Emb}_{\text{speech},1}^{r}(s_{i,1}^{r}) + \sum_{j=2}^{8} \text{Emb}_{\text{speech},j}^{r}(s_{i-1,j}^{r})$, where $s_{i,j}^{r}$ is the $j$-th layer audio token of role $r$ at time $i$, and $\text{Emb}_{\text{speech},j}^{r}$ is the corresponding embedding table.

- **LLM-based retrieval**: An LLM is prompted to read the conversation context and generate a concise, factual reference directly helpful to Moshi's next response, while avoiding non-readable content or formatting. The prompts used are shown in Table 15 in the appendix.
- **Search-based retrieval**: We use the AI-optimized search tool Tavily[6] to access real-time information from the web and extract key highlights as the reference document. While there exist other search tools such as Perplexity, we choose Tavily as it provides a concise summarized output, reducing the need to post-process retrieved information.

As our goal is to make Moshi$_{AG}^{R}$ capable of handling questions across diverse domains, we intentionally adopt general-purpose tools rather than standard RAG databases commonly used in research literature.

## 4. Data and Training

### 4.1. Data Generation

Training of Moshi$_{AG}^{R}$ relies on synthetic data. We first produce text-based conversational scripts on specified topics along with associated reference documents using LLMs. These scripts are then converted into spoken conversations using a dual-speaker conversational TTS system. An overview of the data generation process is provided below.

#### 4.1.1. TOPICS

To construct conversations involving knowledge-intensive queries, we curate a set of topics from existing question-answering datasets. We extract approximately 307k topics from the training split of Natural Questions (Kwiatkowski et al., 2019), 90k topics from HotpotQA (Yang et al., 2018), and 76k topics from TriviaQA (Joshi et al., 2017), for a total of 474k QA-derived topics. In addition to QA-dataset-based topics, we use LLMs to generate conversation topics in specific expert domains. Through iterative discussions with LLMs, we identify 111 expert domains across 16 large categories. We then employ a Gemma 3 27B model (Gemma Team et al., 2024) to generate 50 conversation topics for each domain, resulting in additional 5.5k LLM-generated expert-domain topics. Details on the topic taxonomy and generation process are provided in Appendix D.

#### 4.1.2. CONVERSATION SCRIPTS WITH REFERENCES

For each topic, we use LLMs to generate multi-turn conversational scripts that simulate natural human–assistant interactions. Each script is accompanied by reference documents that support knowledge-intensive responses. An example can be found in the appendix (Table 14).

In these scripts, knowledge-intensive turns by Moshi are

explicitly associated with reference documents to simulate the RAG process. An RAG-enabled response consists of three segments: a **lead** portion that does not depend on external knowledge, a **body** portion that contains reference-grounded content, and an optional **tail** portion that concludes the response. This structure is designed such that, at training time, reference information can be injected before the body segment begins, mirroring the asynchronous RAG mechanism described in Section 3.2.

To generate these scripts, we employ three Gemma 3 27B LLMs, each assigned a distinct role: a *user* LLM, a *Moshi* LLM, and a *reference* LLM. The user LLM has access to the topic and prior conversation context but not the reference documents to prevent information leakage. In contrast, the Moshi and reference LLMs do not observe the topic directly but have full access to the prior conversation context and the reference information. As a result, any topic awareness must be inferred from the user's utterances, as in real human-assistant interaction. When generating conversation scripts, we maintain a universal record similar to the format in Table 14 and generate the scripts line by line. Based on the content that has been generated, we call the appropriate LLM, organize the record into the format for that LLM (e.g. removing references for the user LLM), and then ask the LLM to fill in the next line in the conversation script, until the user LLM decides to end the conversation.

To increase diversity and improve robustness, we design three prompt variants to elicit different interaction styles:

- **v1**: a basic conversation centered on the selected topic.
- **v2**: a conversation where the user challenges Moshi more frequently and therefore more back-and-forth exchanges and argumentation are included.
- **v3**: a conversation where the user occasionally introduces irrelevant remarks or engages in small talk.

For each prompt variant, we generate one multi-turn conversation for each topic and append a Moshi greeting message to the beginning of the script.[7] In addition, we construct a **single-turn** subset for QA-style topics, consisting of a single user question (we directly use the questions from the QA datasets), an LLM-generated reference document, and an LLM-generated Moshi response. These sources combine to form approximately 1.9M conversation instances. Training and validation data statistics are reported in Tables 4 and 5, and example LLM prompts for the v1 setting are provided in Table 15.

---

[7]Some benchmarks adopt a user-first setup (Li et al., 2025; Lin et al., 2025; 2026) where the user is assumed to talk first. To make our model robust to different situations, during training, we remove this greeting message with probability 0.3.

---

[6]https://www.tavily.com

### 4.1.3. SPEECH SYNTHESIS

We employ a multi-channel text-to-speech (TTS) model, similar to the one used to generate the instruction-tuning data in the original Moshi paper (Défossez et al., 2024), to convert the scripts into audio. Following the original Moshi setup, we use a fixed speaker as Moshi's voice and randomly sample another speaker from an internal dataset as the user's voice. The synthesized speech corpus has an average duration of approximately 2 minutes for multi-turn conversations and 15 seconds for the single-turn subset.

### 4.2. Training

Our data generation pipeline produces spoken conversations between two speakers, along with reference documents associated with specific Moshi turns. However, when training the Moshi model, both the timing of the $\langle\text{ret}\rangle$ prediction and the duration of the retrieval process (i.e. the $i_{\text{ret}}$ parameters and the retrieval delay $d$ in Equation 2) are unknown. To place the $\langle\text{ret}\rangle$ token, we leverage the forced alignment between speech and text provided by the multi-channel TTS model. Specifically, we replace the text token before the first text token in the lead portion of an RAG-enabled Moshi turn with the $\langle\text{ret}\rangle$ token. For the retrieval delay, we simulate its value with the following sampling strategy:

$$ d' = \begin{cases} \mathcal{U}(0, d_{\text{lead}}), & \text{if } d_{\text{lead}} < 2 \text{ or } p < 0.2, \\ \mathcal{U}(1.0, d_{\text{lead}} - 1.0), & \text{otherwise.} \end{cases} $$
(3)

where $d'$ denotes the simulated retrieval delay used during training, $d_{\text{lead}}$ is the duration of the lead portion, and $p \sim \mathcal{U}(0,1)$ is a random variable. This design ensures that, in most cases, the retrieval delay is sampled from the interval $(1.0, d_{\text{lead}} - 1.0)$, thereby guaranteeing at least 1.0 second of buffer time before key information in the body portion is mentioned. Meanwhile, the fallback probability 0.2 broadens the distribution to cover edge cases in which retrieval is unusually fast or slow.

Figure 5 illustrates the distributions of retrieval delay during training and inference. While the two distributions overlap, the broader training-time distribution exposes the model to edge cases that potentially enhance robustness. It also shows that inference-time retrieval delays are almost always shorter than the keyword delay, confirming that the timing constraint described in Section 3.1 is almost always satisfied.

We initialize Moshi$_{\text{AG}}^{\text{R}}$ with the original Moshi and make all parameters trainable except for the reference text encoder. A dropout probability of 0.2 is applied to each reference document. When a reference document is dropped, we set the embedding in Equation 2 to $h'_i = h_i + h_{\text{dropout}}$ for $i = i_{\langle\text{ret}\rangle} + \frac{d}{f_r}$, where $h_{\text{dropout}}$ is a learnable vector. We apply window-based filtering to the raw audio signals using an 80ms window size. Audio segments with a

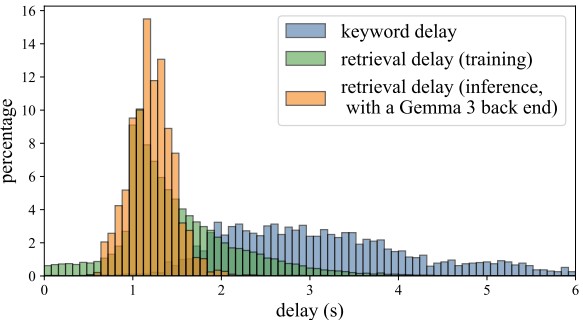

*Figure 5.* Histograms of the distributions of retrieval and keyword delays. Our training data covers a wider range of retrieval delays than observed in practice at inference. Keyword delay is almost always longer than practical retrieval delay, leaving sufficient time to integrate retrieved information to improve the model's answer.

root-mean-square level below $-65$dBFS are zeroed out. The Moshi model is trained on the synthetic dataset for 100k updates, with a learning rate of $2 \times 10^{-6}$ and a batch size of 32. Other training details follow Défossez et al. (2024).

## 5. Experiments

Unless otherwise specified, we use a Gemma 3 27B model with the same prompt as in the data generation phase (see Table 15) as the retrieval back end. We note that existing benchmarks primarily focus on single-turn cases, and do not cover Moshi$_{\text{AG}}^{\text{R}}$'s ability for multi-turn conversations.

### 5.1. Factuality

We evaluate factuality using speech QA datasets. Each dataset consists of spoken questions paired with ground-truth textual answers. We use audio files from OpenAudioBench (Li et al., 2025) and follow its evaluation protocol, using LLM judges to assess the correctness of Moshi$_{\text{AG}}^{\text{R}}$'s responses on the Llama Questions, Web Questions, and TriviaQA datasets. In addition, we construct a more challenging evaluation set based on the HaluEval corpus.[8][9] We gather the first 1,000 instances from the "qa" subset of HaluEval and synthesize spoken questions using our multi-channel TTS model, with user voices randomly sampled from the CommonVoice (Ardila et al., 2020) corpus. Results for Moshi$_{\text{AG}}^{\text{R}}$ and competing speech LMs are summarized in Table 1.

We observe a substantial performance gain from Moshi to Moshi$_{\text{AG}}^{\text{R}}$, especially on the more challenging benchmarks. This improvement can be largely attributed to the integration

---

[8]The HaluEval (Li et al., 2023) dataset provides ground-truth reference documents, suited for evaluating RAG methods. Ablation studies with those are reported in Appendix B.2.

[9]https://huggingface.co/datasets/kyutai/HaluEvalAudio_1000

*Table 1.* Results for factuality evaluation, delay metrics, and computation consumption of various models. Darker color indicates better performance. Underlined numbers are values reported in the original paper of each model. "-" indicates values inaccessible since part or all of the models are not released. "ref." and "resp." show the correctness of retrieved reference and Moshi$_\text{G}^\text{R}$'s final response, respectively.

| Model (Base LM Size) | Accuracy (%) | | | | Delay (sec.) | | | Computation |
| --- | --- | --- | --- | --- | --- | --- | --- | --- |
| | LlamaQ ref.\|resp. | WebQ ref.\|resp. | TriviaQA ref.\|resp. | HaluEval ref.\|resp. | TTFAT | KD | E2EKD | FLOPs/sec. |
| GPT-4o Audio (OpenAI, 2024) | 88.4 | 81.0 | 90.6 | 68.7 | - | 5.5 | - | - |
| GLM-4-Voice (9B) (Zeng et al., 2024) | 64.7 | 32.2 | 39.1 | 21.2 | 0.3 | 4.2 | 4.4 | 0.74 |
| Freeze-Omni (7B) (Wang et al., 2025c) | 72.0 | 44.7 | 53.9 | 14.0 | 1.2 | 5.2 | 6.5 | 0.18 |
| MinMo (7B) (Chen et al., 2025a) | 78.9 | 55.0 | 48.3 | - | - | - | - | - |
| LUCY (7B) (Gao et al., 2025) | 59.7 | 29.3 | 27.0 | | - | - | - | - |
| Step-Audio-Chat (130B) (Huang et al., 2025) | 81.0 | 75.1 | 58.0 | 21.0 | 6.2 | 4.3 | 10.4 | 5.03 |
| Baichuan-Audio (7B) (Li et al., 2025) | 78.4 | 64.5 | 61.7 | 25.0 | 1.0 | 3.8 | 4.8 | 0.84 |
| LLaMA-Omni2 (14B) (Fang et al., 2025b) | 73.0 | 40.4 | 63.8 | 28.8 | 0.1 | 2.8 | 2.9 | 2.58 |
| Qwen 2.5 Omni (7B) (Xu et al., 2025a) | 79.0 | 62.0 | 58.0 | 33.7 | 1.1 | 3.2 | 4.3 | 4.57 |
| Kimi-Audio (7B) (KimiTeam et al., 2025) | 79.3 | 70.2 | 62.1 | 43.2 | 0.2 | 3.3 | 3.5 | 6.93 |
| SALMONN-omni (8B) (Yu et al., 2025) | 80.0 | 50.5 | 66.0 | - | - | - | - | - |
| STITCH-S (9B) (Chiang et al., 2026) | 73.3 | 50.2 | 50.0 | - | - | - | - | - |
| Qwen3-Omni-A3B-Ins. (30B) (Xu et al., 2025b) | 84.7 | 68.8 | 73.6 | 38.9 | 3.7 | 2.0 | 5.7 | 0.57 |
| MoshiRAG$_{\text{Gemma 3 27B}}$ (7B)* | 83.0 \|80.3 | 71.5 \|67.2 | 73.7 \|69.6 | 42.0 \|36.3 | 0.0 | 3.1 | 3.1 | 0.37 |
| MoshiRAG$_{\text{GPT 4.1}}$ (7B) | 87.8 \|80.6 | 77.7 \|68.9 | 86.8 \|78.2 | 61.2 \|51.3 | | | | - |
| MoshiRAG$_{\text{Tavily}}$ (7B) | 84.6 \|78.2 | 73.5 \|66.1 | 84.9 \|77.5 | 54.3 \|47.0 | | | | - |
| Vanilla Moshi (7B) (Défossez et al., 2024) | 62.3 | 26.6 | 22.8 | 10.5 | 0.0 | 2.1 | 2.1 | 0.22 |
| Vanilla Moshi fine-tuned on RAG data (7B) | 61.2 | 37.0 | 29.7 | 18.7 | 0.0 | 3.1 | 3.1 | 0.22 |

*Moshi$_\text{A}^\text{R}$ with a Gemma 3 27B back end is used as the default method throughout the paper if not otherwise specified.

of RAG, as Moshi$_\text{G}^\text{R}$ also significantly outperforms a vanilla Moshi model fine-tuned on the RAG training data. Overall, Moshi$_\text{G}^\text{R}$ achieves performance that is comparable to, and in many cases better than, most existing speech LMs (most of which are not full-duplex), except for GPT-4o Audio.[10]

The "ref." columns in Table 1 report the accuracy of the retrieved reference information provided by the back ends. On average, these scores are about 5% higher than the accuracy of Moshi$_\text{G}^\text{R}$'s final spoken responses. This gap reflects information loss introduced during RAG integration and highlights opportunities for further improvement. On the other hand, the ref. score also marks a performance upper bound for Moshi$_\text{G}^\text{R}$. Fortunately, this upper bound can be further improved by switching to a more powerful knowledge source. When GPT-4.1 or Tavily search is used as the retrieval back end,[11] Moshi$_\text{G}^\text{R}$ achieves substantial gains on the more challenging TriviaQA and HaluEval datasets and outperforms all compared speech LMs except GPT-4o Audio.

## 5.2. Delay and Computation Consumption

Beyond factuality, the time and computation required to fulfill users' requests are also critical factors for speech LMs. We measure the TTFAT, as defined in Section 3.1, for publicly available models by recording the time elapsed until the first audio token is emitted when running each

model on a single H100 GPU.[12] To compute keyword delay, we use a Gemma 3 27B model to extract the keyword from each model response (prompt provided in Table 17 in the appendix). We then obtain the onset time of the keyword using timestamps marked by the `parakeet-tdt-0.6b-v2` ASR model.[13] Computation cost is estimated using the flops profiler in the DeepSpeed package,[14] and the average number of floating-point operations (FLOPs) required to generate one second of audio is reported. All metrics are macro-averaged across datasets and shown in Table 1.

Compared to the vanilla Moshi model, the conversational template used by Moshi$_\text{G}^\text{R}$ (i.e., the lead portion in RAG-enabled turns) introduces a one-second increase in keyword delay. However, the E2EKD of Moshi$_\text{G}^\text{R}$ is lower than that of nearly all competing systems. When accounting for retrieval overhead, Moshi$_\text{G}^\text{R}$'s computation cost remains comparable to other models of similar scale. These results demonstrate that Moshi$_\text{G}^\text{R}$ achieves strong factual performance with reasonable trade-offs in delay and computational efficiency.

## 5.3. Interactivity

Evaluating the interactivity between voice assistants and human users has long been a challenging open problem. We assess this aspect using Full-Duplex-Bench (Lin et al.,

---

[10]We use checkpoint `gpt-4o-audio-preview`.

[11]Due to unstable API response times, we assume a uniform retrieval delay of 1.5 sec. – higher than 90% of cases with a local Gemma model – for all experiments with non-Gemma back ends.

[12]Exceptions are Moshi$_\text{G}^\text{R}$ and Step-Audio-Chat. For Moshi$_\text{G}^\text{R}$, the front-end models run on one GPU, and the local retrieval back end runs on another. Step-Audio-Chat runs on four GPUs.

[13]https://huggingface.co/nvidia/parakeet-tdt-0.6b-v2

[14]https://www.deepspeed.ai/tutorials/flops-profiler

*Table 2.* Evaluation results on Full-Duplex-Bench (Lin et al., 2025). Underlined numbers are values reported in the original paper of each model. $TOR_s$ and $TOR_c$ in the Pause track correspond to the synthetic subset and the Candor subset, respectively.

| Model | Pause | | Backchannel | | | Turn Taking | | User Interruption | | |
|---|---|---|---|---|---|---|---|---|---|---|
| | $TOR_s \downarrow$ | $TOR_c \downarrow$ | TOR $\downarrow$ | Freq. (/s) $\uparrow$ | JSD $\downarrow$ | TOR $\uparrow$ | Latency (s) $\downarrow$ | TOR $\uparrow$ | GPT Score $\uparrow$ | Latency (s) $\downarrow$ |
| dGSLM (Nguyen et al., 2023) | 0.93 | 0.94 | 0.69 | 0.015 | 0.93 | 0.98 | 0.35 | 0.92 | 0.20 | 2.53 |
| Freeze-Omni | 0.64 | 0.48 | 0.64 | 0.001 | 1.00 | 0.34 | 0.95 | 0.87 | 3.62 | 1.41 |
| Gemini (Comanici et al., 2025) | 0.26 | 0.31 | 0.09 | 0.012 | 0.90 | 0.66 | 1.30 | 0.89 | 3.38 | 1.18 |
| MoshiRAG | 0.32 | 0.56 | 0.64 | 0.010 | 0.94 | 0.83 | 0.18 | 0.85 | 3.75 | 1.02 |
| Vanilla Moshi | 0.99 | 0.98 | 1.00 | 0.001 | 0.96 | 0.94 | 0.27 | 1.00 | 0.77 | 0.26 |
| Vanilla Moshi fine-tuned | 0.39 | 0.63 | 0.64 | 0.017 | 0.91 | 0.98 | 0.18 | 0.95 | 4.19 | 0.52 |

2025), which evaluates model behavior in specific conversational scenarios. The benchmark consists of pre-recorded user audio and evaluates model reactions to the audio input, with particular emphasis on turn-taking behavior. Specifically, the **pause** track measures whether a model refrains from taking the turn before the user has finished speaking; lower takeover rates (TOR) are therefore preferred. The **backchannel** track evaluates both the backchanneling frequency per second during user speech and the similarity of their temporal distribution to the ground-truth human-to-human conversation, measured by Jensen–Shannon divergence (JSD). Although there is no consensus on the optimal amount of backchanneling, the benchmark favors higher backchannel frequency, as many existing speech LMs rarely backchannel. The **turn taking** track assesses whether the model takes the turn promptly after the user finishes speaking. Finally, the **user interruption** track uses 5-scale GPT score to evaluate how well the model responds to user interruptions during its own speech and tests whether the model can successfully and smoothly resume the conversation after the interruption without excessive delay.

As shown in Table 2, $Moshi_{G}^{R}$ consistently exhibits lower TORs than the original Moshi across all evaluated conditions. As a Moshi model fine-tuned on RAG data demonstrates similar behavior, this effect can largely be attributed to the training data distribution: longer, knowledge-intensive turns result in more conservative turn-taking and therefore reduced TOR. At the same time, $Moshi_{G}^{R}$ maintains consistently lower latency than Freeze-Omni and Gemini, preserving the real-time interaction advantages of full-duplex systems that the original Moshi also exhibits. Furthermore, both $Moshi_{G}^{R}$ and the RAG-fine-tuned Moshi respond significantly better to user interruptions than the original Moshi model. This improvement is likely derived from the v2 and v3 training subsets that expose the model to adversarial and distracting conversational scenarios, enabling the model to rapidly adapt to changing conversational topics and contexts and to promptly address the user's most recent requests.

*Table 3.* Evaluation results on out-of-domain mathematical reasoning datasets. Underlined numbers are values reported in the STITCH paper (Chiang et al., 2026). $MoshiRAG_{summ.}$ means that the reference is summarized before being injected into Moshi.

| Model | Accuracy (%) | | | | | | | | | |
|---|---|---|---|---|---|---|---|---|---|---|
| | AddSub | | MultiArith | | SinglEq | | SVAMP | | GSM8K | |
| | ref. | resp. | ref. | resp. | ref. | resp. | ref. | resp. | ref. | resp. |
| GLM-4-Voice | 59.4 | | 62.0 | | 71.0 | | 4.0 | | 29.0 | |
| STITCH-S | 81.7 | | 87.9 | | 91.7 | | 72.2 | | 56.7 | |
| MoshiRAG | 76.6 | 61.7 | 87.1 | 69.0 | 83.2 | 68.2 | 74.1 | 55.0 | 66.2 | 33.9 |
| $MoshiRAG_{summ.}$ | 62.0 | | 73.1 | | 66.4 | | 57.5 | | 51.2 | |
| $MoshiRAG_{GPT 4.1}$ | 87.9 | 64.8 | 87.1 | 76.0 | 89.6 | 72.9 | 80.5 | 61.1 | 70.8 | 43.2 |
| Vanilla Moshi | 8.3 | | 9.8 | | 18.4 | | 9.7 | | 2.1 | |

### 5.4. Generalization to Unseen Tasks

Although $Moshi_{G}^{R}$ is trained primarily on QA-style data, the system design enables $Moshi_{G}^{R}$ to generalize to out-of-distribution queries. As long as the retrieval back end can successfully handle a question, $Moshi_{G}^{R}$ can leverage retrieval as an external tool to extend its capabilities beyond the training distribution. To validate this hypothesis, we adopt the mathematical reasoning datasets used by STITCH (Chiang et al., 2026), a speech LM specialized for math reasoning, and generate spoken questions following the same procedure used for HaluEval. The results are shown in Table 3. While $Moshi_{G}^{R}$ does not yet match models explicitly trained for mathematical reasoning, it substantially outperforms non-reasoning speech LMs such as GLM-4-Voice and the vanilla Moshi, demonstrating meaningful generalization beyond QA-centric tasks.

Interestingly, we observe that, rather than directly using the LLM-generated reference, additionally instructing the LLM to summarize the reference before knowledge integration leads to improved performance in Table 3. We hypothesize that, despite prompt instructions to avoid it, the initial LLM-generated references often contain excessive numerical details, symbolic expressions, and lengthy reasoning processes, which make the knowledge integration less effective, as evidenced by the large gap between the ref. and the resp. scores. Compressing this content allows $Moshi_{G}^{R}$ to focus on the core concepts and leads to more accurate responses.

# 6. Conclusion

We propose Moshi$_{AG}^{R}$, the first attempt to integrate RAG into a full-duplex speech language model. Our system enables Moshi to trigger an asynchronous retrieval process when encountering knowledge-demanding user queries, while allowing the conversation between the user and the model to continue uninterrupted. Leveraging the natural temporal gap between the user's query and the model's delivery of core information, the retrieval process can obtain supporting evidence either from a more knowledgeable LLM or via web search, and ground Moshi's response in factual references. This approach significantly improves factuality, outperforming most publicly released turn-based speech language models, while preserving the high interactivity inherent to full-duplex systems. Experimental results also demonstrate strong performance on mathematical reasoning tasks that Moshi$_{AG}^{R}$ has not been explicitly trained on, showing out-of-domain generalization of its tool use ability.

Currently, the retrieval trigger in Moshi$_{AG}^{R}$ relies entirely on training data. In future work, we aim to improve this by linking retrieval decisions to query difficulty, or by applying reinforcement learning to determine whether retrieval is necessary. We plan to diversify the set of retrieval tools and enable the model to select appropriate tools based on user input. Meanwhile, improving the robustness of the model itself remains crucial, particularly to improve its robustness against errors that may occur during the retrieval process.

## Acknowledgements

The authors wish to acknowledge Hippolyte Pilchen for his contributions in incorporating ARC-Encoder into this project, and Gabriel de Marmiesse for his support in the Moshi$_{AG}^{R}$ demo.

## Impact Statement

This work advances research on speech language models by enhancing factuality and reliability without compromising interactivity in full-duplex voice interactions through asynchronous retrieval-augmented generation. By allowing full-duplex models to access external knowledge sources, the proposed approach enables more helpful, accurate, and natural voice-based conversations. The improvements have particularly strong implications for accessibility, benefiting users who rely on voice interfaces as a primary means of obtaining information.

At the same time, increased conversational realism and stronger factual grounding may heighten user trust in voice assistants, underscoring the importance of incorporating appropriate safeguards against misinformation, over-reliance, and misuse. As with other retrieval-augmented systems, inaccuracies or biases in retrieved sources may be propagated in model outputs. Addressing these risks requires careful system design, transparent deployment practices, and ongoing evaluation. While these considerations fall outside the scope of this work, they represent important directions for future research.

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

## A. Data Statistics

Table 4 shows the statistics of the synthetic training data, generated following the pipeline described in Section 4.1. We employ a Gemma 3 27B LLM to generate 5.5k conversation topics and extract 472k questions from the training sets of the QA datasets HotpotQA ("fullwiki" split), Natural Questions, and TriviaQA ("rc" split) as the QA-based topics. Using these topics, we synthesize multi-turn conversation scripts with the v1, v2, and v3 prompts, and then convert the scripts to speech using our multi-channel TTS model. After removing failed generation samples, this process produces approximately 475k conversations per subset. Conversation lengths vary across subsets, with v1 averaging 1.5 minutes per conversation and v3 averaging 2.5 minutes approximately.

In addition, we create a single-turn subset using the QA datasets. Here, the questions in the datasets are used as the user's query, while a *reference* LLM generates the reference document and a *Moshi* LLM produces Moshi's response. This subset is similar to the format of the QA benchmarks used in Section 5, which exposes Moshi with examples where the user directly asks a question without any greeting – the multi-turn subsets always start with greetings.

The statistics of the validation data are shown in Table 5. Similar to the training data, we use the validation sets of the QA datasets to generate both multi-turn and single-turn conversations. Unlike the training data, LLM-generated topics are not used for validation.

*Table 4.* Moshi$_{\text{G}}^{\text{R}}$ A training data statistics. Values are reported as: **number of conversations (hours)**.

| Subset | LLM (# conv / hrs) | HotpotQA (# conv / hrs) | NQ (# conv / hrs) | TriviaQA (# conv / hrs) | Total (# conv / hrs) |
|---|---|---|---|---|---|
| Multi-turn v1 | 5457 (231) | 90208 (2201) | 305077 (8083) | 76068 (1866) | 476810 (12381) |
| Multi-turn v2 | 5549 (236) | 90034 (2433) | 303359 (9264) | 75515 (2097) | 474457 (14030) |
| Multi-turn v3 | 5550 (298) | 90208 (3459) | 306754 (12701) | 76061 (2947) | 478573 (19405) |
| Single-turn | | 90172 (438) | 305364 (333) | 471536 (1954) | |
| Total | (765) | (8531) | (31231) | (7243) | 1901376 (47770) |

*Table 5.* Moshi$_{\text{G}}^{\text{R}}$ A validation data statistics. Values are reported as: **number of conversations (hours)**.

| Subset | HotpotQA (# conv / hrs) | NQ (# conv / hrs) | TriviaQA (# conv / hrs) |
|---|---|---|---|
| Multi-turn v1 | 7403 (181) | 7830 (208) | 9956 (245) |
| Multi-turn v2 | 7391 (199) | 7814 (240) | 9939 (271) |
| Multi-turn v3 | 7403 (283) | 7826 (324) | 9954 (385) |
| Single-turn | 7403 (15) | 7830 (10) | 9956 (18) |
| Total | (678) | (882) | (919) |

## B. Experiments

### B.1. Justification of Model Architecture

In the early state of our development, we experiment with several different model architectures, primarily focusing on the reference text encoding and information injection modules. For reference encoding, we evaluate T5 (Raffel et al., 2020) and ARC-Encoder (Pilchen et al., 2025). As ARC-Encoder is explicitly designed for sequence length compression, the key difference between the two options is the length of their output sequences. This difference is critical for Moshi$_{\text{A}}^{\text{R}}$ G due to its strict timing constraints – shorter encoded sequences allow the Moshi model to consume reference information earlier and more efficiently during streaming inference. For information injection, we consider two strategies: additive injection and insertive injection.[15] Additive injection adds the reference embedding to Moshi's input embedding in a streaming manner, as defined in Equation 2, so the input sequence length remains unchanged. In contrast, insertive injection explicitly inserts the reference embedding sequence into Moshi's input sequence, thereby increasing the total sequence length.

---

[15]Initially, we also experiment with cross-attention-based injection, but the training with cross-attention does not converge as good as the insertive approach so we omit cross-attention from the comparison here to simplify the experimental setup and avoid additional architectural modifications required.

### B.1.1. INFORMATION INJECTION

We construct a modified training set for the comparison of different information injection strategies. In this dataset, all references throughout the entire conversation are condensed into a single passage to reduce the total reference length.[16] The primary goal of this experiment is to study the injection strategy. Therefore, to eliminate confounding factors, we use ground-truth user transcriptions instead of ASR predictions, and inject reference information at the beginning of the conversation rather than following inference-time retrieval delays. The results are reported in Table 6.

For both reference encoders, insertive injection consistently outperforms additive injection, demonstrating a clear trade-off between information integration effectiveness and Moshi's input sequence length. This performance gap is larger when using T5 than ARC-Encoder. We attribute this behavior to the excessive length of T5's output – with additive injection, reference information could be introduced too late for Moshi to effectively utilize it. This result further validates the importance of precise timing control in the $\text{Moshi}_{\text{A}\text{G}}^{\text{R}}$ framework.

Despite the better performance of insertive injection, we ultimately adopt the additive strategy to constrain sequence length, which preserves Moshi's ability to sustain long-form conversations. Nevertheless, designing an information injection mechanism that jointly optimizes performance and efficiency remains an open research question to be further explored.

*Table 6.* Performance of $\text{Moshi}_{\text{A}\text{G}}^{\text{R}}$ with different information injection strategies. Results are scored with a Gemma 3 27B LLM. The default $\text{Moshi}_{\text{A}\text{G}}^{\text{R}}$ setup adopts ARC-Encoder with additive injection.

| Training Data | Ref. Encoder | Injection | Accuracy (%) | | | |
|---|---|---|---|---|---|---|
| | | | LlamaQ | WebQ | TriviaQA | HaluEval |
| Single-ref. | T5 | Insertive | 81.0 | 79.0 | 82.0 | 47.1 |
| | | Additive | 59.6 | 36.6 | 28.5 | 13.9 |
| Single-ref. | ARC-Encoder | Insertive | 80.7 | 78.6 | 82.2 | 49.0 |
| | | Additive | 76.4 | 67.4 | 73.1 | 41.8 |

### B.1.2. REFERENCE ENCODER

Next, we evaluate the choice of reference encoder by comparing T5 with two ARC-Encoder variants with compression ratios of four and eight. Unlike the controlled setting in Section B.1.1, we revert to the default experimental setup with multi-turn training data, ASR-predicted user transcriptions, and additive injection aligned with inference-time retrieval delays. Given the sensitivity of additive injection to sequence length when using T5, we additionally experiment with a pre-encoding summarization strategy that summarizes reference text prior to encoding, using the prompt shown in Table 15.

As shown in Table 7, ARC-Encoder with a compression ratio of four consistently outperforms the other configurations. While pre-encoding summarization improves performance for T5, it is not as useful for ARC-Encoder. Based on these results, we adopt ARC-Encoder with a compression ratio of four, without pre-encoding summarization, and use additive injection as the default setup throughout the paper. This configuration corresponds exactly to the system design described in Section 3.2.

*Table 7.* Performance of $\text{Moshi}_{\text{A}\text{G}}^{\text{R}}$ with different reference text encoders. Results are scored with a Gemma 3 27B LLM. The default $\text{Moshi}_{\text{A}\text{G}}^{\text{R}}$ setup adopts ARC-Encoder with a compression ratio of four without pre-encoding summarization.

| Summarization | Ref. Encoder | Injection | Accuracy (%) | | | |
|---|---|---|---|---|---|---|
| | | | LlamaQ | WebQ | TriviaQA | HaluEval |
| ✗ | ARC-8 | Additive | 77.4 | 66.3 | 66.2 | 33.1 |
| | ARC-4 | Additive | 80.3 | 74.7 | 73.2 | 36.3 |
| | T5 | Additive | 76.5 | 70.9 | 72.1 | 31.7 |
| ✓ | ARC-4 | Additive | 77.4 | 75.4 | 73.5 | 37.5 |
| | T5 | Additive | 80.5 | 71.4 | 71.5 | 32.9 |

---

[16]Otherwise, the excessive Moshi input sequence length in the "insertive" setting makes training infeasible under our computation constraints.

## B.2. Sensitivity to ASR and Reference Correctness

Moshi$_{AG}^{R}$ relies on the retrieval back end to provide factual reference information, while the information retrieval relies on the outputs of Moshi and the streaming ASR model to provide accurate conversation context. Errors introduced at any stage of this pipeline can propagate and accumulate, ultimately affecting the quality of Moshi$_{G}^{R}$'s final responses. To analyze the impact of such cumulative errors, we evaluate Moshi$_{AG}^{R}$'s performance on QA benchmarks using ground-truth user transcriptions and its performance on HaluEval when ground-truth reference documents are provided. As shown in Table 8, Moshi$_{AG}^{R}$ is highly sensitive to ASR errors. Improving ASR accuracy could substantially improve the correctness of retrieved references and final responses by up to 15%. Meanwhile, providing ground-truth reference documents also leads to an improvement in the response accuracy. However, the gap between reference and response accuracies also increases, reflecting significant information loss during the knowledge integration process. This experiment highlights two straightforward directions for improving Moshi$_{AG}^{R}$'s factuality: more accurate modeling of conversation context and more effective integration of retrieved information.

*Table 8.* Ablation study on QA benchmarks using ground-truth user text and ground-truth reference. Results are scored with a Gemma 3 27B LLM. Despite the 3 to 6 % correctness loss in the knowledge integration process (the gaps between ref. and resp. columns), the significantly improved performance when using ground-truth transcriptions demonstrates the potential of improving Moshi$_{AG}^{R}$ by more accurate context modeling. However, the enlarged gap between reference and response accuracies when using HaluEval's ground-truth reference also indicates that the knowledge integration process can be further improved.

| User Text | Reference | \\ Accuracy (%) | | | | | | | |
| | | LlamaQ | | WebQ | | TriviaQA | | HaluEval | |
| | | ref. | resp. | ref. | resp. | ref. | resp. | ref. | resp. |
| --- | --- | --- | --- | --- | --- | --- | --- | --- | --- |
| ASR | Gemma LLM | 83.3 | 80.3 | 79.3 | 74.7 | 76.9 | 73.2 | 42.0 | 36.3 |
| ground-truth | Gemma LLM | 84.8 | 80.2 | 84.7 | 78.3 | 85.8 | 82.5 | 57.2 | 50.8 |
| - | ground-truth | | | | | | | 97.2 | 65.1 |
| ASR Word Error Rate (%) | | 0.7 | | 5.2 | | 4.6 | | 11.5 | |

## B.3. Experiment of Diverse Retrieval Back Ends

To further evaluate the architectural flexibility of Moshi$_{AG}^{R}$, we extend the primary evaluation in Section 5 beyond the initial Gemma 3, GPT-4.1, and Tavily retrieval back ends. In this section, we examine Moshi$_{AG}^{R}$'s performance with a range of LLM-based back ends, spanning edge-capable models to large-scale frontier LLMs. We select three representative models covering a broad spectrum of parameter scales: Llama 4 Maverick,[17] Mistral Medium 3.1,[18] and Gemini 2.5 Flash (Comanici et al., 2025). Results for the QA and mathematical reasoning benchmarks are reported in Tables 9 and 10, respectively.

Despite only being trained on synthetic data generated using Gemma 3, Moshi$_{AG}^{R}$ exhibits strong cross-model stability. Performance remains largely consistent when paired with GPT-4.1, Mistral, or Gemini back ends, suggesting that the RAG framework is largely agnostic to the linguistic characteristics of the underlying retriever. This stability across different back ends provides important flexibility for real-world system design. While premium models such as GPT-4.1 establish an upper bound on performance, smaller LLMs like Gemini Flash achieve competitive baseline results at substantially lower inference costs. The strong performance of these smaller models indicates that Moshi$_{AG}^{R}$ may be suitable for deployment in resource-constrained or edge environments, reducing reliance on external APIs. Finally, support for multiple back ends enables the implementation of redundancy mechanisms, such as using a local small LLM as a fallback strategy for online APIs, which potentially improves Moshi$_{AG}^{R}$'s resilience to API outages or retrieval failures.

## B.4. Results on Full-Duplex-Bench v1.5

To further study Moshi$_{AG}^{R}$'s interactivity with users, we evaluate it using the Full-Duplex-Bench v1.5 (Lin et al., 2026) benchmark. This benchmark is designed to assess how a speech LM reacts when it suddenly receives overlapping speech input from the user while it is speaking. Four types of overlapping speech are considered: the user interrupts, the user backchannels, the user talks to other people, or non-user speakers talk in the background. The models' reactions are classified into four categories using GPT-4o:

---

[17]https://ai.meta.com/blog/llama-4-multimodal-intelligence
[18]https://docs.mistral.ai/models/mistral-medium-3-1-25-08

*Table 9.* Evaluation of Moshi$_{AG}^{R}$ on QA Benchmarks with different retrieval back ends. Results are scored with a Gemma 3 27B LLM. GPT 4.1 consistently shows the best results while web search powered by Tavily also shows competitive performance.

| Retrieval Back End | Accuracy (%) | | | | | | | |
| --- | --- | --- | --- | --- | --- | --- | --- | --- |
| | LlamaQ | | WebQ | | TriviaQA | | HaluEval | |
| | ref. | resp. | ref. | resp. | ref. | resp. | ref. | resp. |
| Gemma | 83.3 | 80.3 | 79.3 | 74.7 | 76.9 | 73.2 | 42.0 | 36.3 |
| GPT 4.1 | 86.6 | 80.6 | 81.7 | 75.4 | 88.4 | 82.9 | 61.2 | 51.3 |
| Llama 4 Maverick | 81.3 | 78.1 | 79.3 | 72.9 | 81.6 | 70.5 | 45.0 | 33.8 |
| Mistral Medium 3.1 | 86.6 | 79.2 | 83.1 | 72.9 | 87.8 | 80.8 | 53.7 | 44.0 |
| Gemini 2.5 Flash | 82.6 | 80.0 | 76.0 | 72.4 | 82.4 | 77.5 | 50.5 | 41.9 |
| Tavily Search | 85.4 | 79.2 | 81.8 | 76.2 | 86.9 | 81.6 | 54.3 | 47.0 |

*Table 10.* Evaluation of Moshi$_{AG}^{R}$ on mathematical reasoning benchmarks with different retrieval back ends. Results are scored with a Gemma 3 27B LLM. GPT 4.1 again performs the best on all tracks. We do not include Llama 4 Maverick in this experiment as its performance on the QA benchmarks does not provide a favorable trade-off considering its model size and inference cost.

| Retrieval Back End | Accuracy (%) | | | | | | | | | |
| --- | --- | --- | --- | --- | --- | --- | --- | --- | --- | --- |
| | AddSub | | MultiArith | | SinglEq | | SVAMP | | GSM8K | |
| | ref. | resp. | ref. | resp. | ref. | resp. | ref. | resp. | ref. | resp. |
| Gemma | 76.6 | 61.7 | 87.1 | 69.0 | 83.2 | 68.2 | 74.1 | 55.0 | 66.2 | 33.9 |
| GPT 4.1 | 87.9 | 64.8 | 87.1 | 76.0 | 89.6 | 72.9 | 80.5 | 61.1 | 70.8 | 43.2 |
| Mistral Medium 3.1 | 82.2 | 63.9 | 84.8 | 71.8 | 84.1 | 72.9 | 73.5 | 57.5 | 60.8 | 38.4 |
| Gemini 2.5 Flash | 84.1 | 63.0 | 88.1 | 67.3 | 87.9 | 67.3 | 75.5 | 53.1 | 66.8 | 38.8 |

- Respond: the model addresses the content of the overlapping speech.

- Resume: the model ignores the overlapping speech and continues speaking.

- Uncertain: the model expresses confusion.

- Unknown: the model produces an irrelevant response or remains silent.

Different reactions are expected depending on the type of overlapping event. The evaluation results are presented in Table 11. Compared to vanilla Moshi, Moshi$_{AG}^{R}$ shows a higher tendency to resume speaking across all types of overlapping events. This aligns with our earlier findings in Section 5.3, where Moshi$_{AG}^{R}$ consistently exhibits lower TOR than vanilla Moshi. Importantly, when the user really interrupts, the percentage that Moshi$_{AG}^{R}$ responses to the user inputs is similar to vanilla Moshi. This mirrors its performance on the "user interruption" track in Table 2, where Moshi$_{AG}^{R}$ successfully addresses most interruptions and achieves high GPT scores, demonstrating the effectiveness of the v2 and v3 training subsets.

*Table 11.* Evaluation results on Full-Duplex-Bench 1.5 (Lin et al., 2026) across different types of overlapping speech. Underlined numbers are values reported in the original benchmark paper.

| Model | User Interruption | | | | User Backchannel | | | | Talking to Other | | | | Background Speech | | | |
| --- | --- | --- | --- | --- | --- | --- | --- | --- | --- | --- | --- | --- | --- | --- | --- | --- |
| | Resp. ↑ | Resm. ↓ | Unc. ↓ | Unk. ↓ | Resp. ↓ | Resm. ↑ | Unc. ↓ | Unk. ↓ | Resp. ↓ | Resm. ↑ | Unc. ↑ | Unk. ↓ | Resp. ↓ | Resm. ↑ | Unc. ↑ | Unk. ↓ |
| Freeze-Omni | 0.72 | 0.12 | 0.03 | 0.13 | 0.07 | 0.80 | 0.02 | 0.11 | 0.58 | 0.25 | 0.00 | 0.15 | 0.63 | 0.25 | 0.01 | 0.11 |
| Gemini | 0.33 | 0.55 | 0.01 | 0.10 | 0.01 | 0.93 | 0.02 | 0.04 | 0.00 | 0.99 | 0.00 | 0.01 | 0.70 | 0.30 | 0.00 | 0.00 |
| Sonic (Amazon AGI et al., 2025) | 0.24 | 0.71 | 0.01 | 0.04 | 0.00 | 0.98 | 0.00 | 0.02 | 0.10 | 0.90 | 0.00 | 0.00 | 0.01 | 0.98 | 0.00 | 0.01 |
| GPT-4o Audio | 0.78 | 0.10 | 0.02 | 0.12 | 0.03 | 0.70 | 0.01 | 0.25 | 0.91 | 0.02 | 0.01 | 0.06 | 0.93 | 0.04 | 0.00 | 0.03 |
| MoshiRAG | 0.51 | 0.35 | 0.00 | 0.15 | 0.05 | 0.61 | 0.00 | 0.34 | 0.17 | 0.58 | 0.02 | 0.23 | 0.29 | 0.49 | 0.00 | 0.22 |
| Vanilla Moshi | 0.50 | 0.26 | 0.00 | 0.25 | 0.02 | 0.06 | 0.00 | 0.92 | 0.20 | 0.19 | 0.02 | 0.59 | 0.21 | 0.07 | 0.01 | 0.71 |
| Vanilla Moshi fine-tuned | 0.68 | 0.20 | 0.02 | 0.11 | 0.05 | 0.72 | 0.00 | 0.22 | 0.22 | 0.58 | 0.02 | 0.18 | 0.13 | 0.54 | 0.02 | 0.31 |

# C. Further Analysis of Moshi$_{AG}^{R}$ Performance

The performance of Moshi$_{AG}^{R}$ can be evaluated across four primary dimensions: (a) whether retrieval is successfully triggered, (b) the correctness of the retrieved content, (c) the computational overhead and time consumption of the retrieval process,

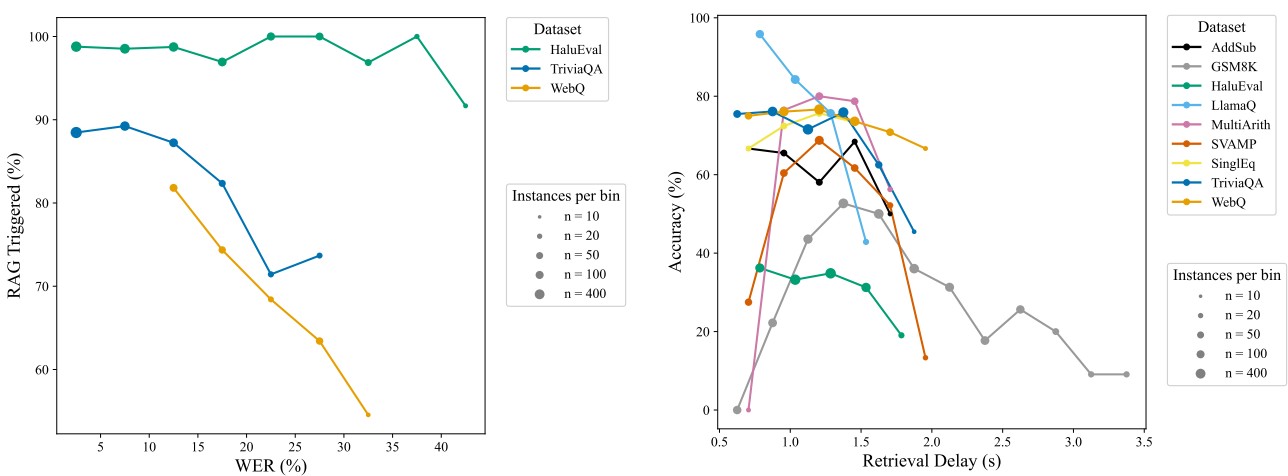

*(a)* RAG trigger rate versus user speech intelligibility (WER). LlamaQ results are excluded as WER remains consistently below 5%.

*(b)* Moshi$_{AG}^{R}$'s response accuracy as a function of retrieval delay for QA datasets and mathematical reasoning datasets.

*Figure 6.* Analysis of Moshi$_{AG}^{R}$ performance under different speech intelligibility and retrieval delay conditions.

and (d) whether the retrieved information is successfully integrated and effectively improves Moshi$_{AG}^{R}$'s final response. Since the results in Table 1 (specifically the ref. and resp. columns) already address retrieval correctness and integration, this section focuses on triggering (a) and time consumption (c) of the retrieval process to provide a more comprehensive view of RAG's effectiveness in full-duplex conversations.

To address the reliability of retrieval triggering, we analyze the relationship between successful RAG triggering and the intelligibility of user speech, as measured by WER. As shown in Figure 6a, while results vary across datasets, RAG is generally triggered more consistently when speech input is clear; trigger rates decline predictably as WER increases. Regarding time consumption, we examine Moshi$_{AG}^{R}$'s performance across datasets relative to retrieval delay. Although Moshi$_{AG}^{R}$ is trained on a broad distribution of retrieval delay values (see Figure 5), we observe a sharp decline in accuracy when retrieval latency exceeds 1.5 seconds across almost all datasets. This highlights that an efficient retrieval backend is critical for good performance. Fortunately, Figure 5 confirms that most inference-time retrieval delays remain below the 1.5-second threshold when utilizing a local Gemma back end, ensuring the stable operation of Moshi$_{AG}^{R}$.

# D. Data Generation Resources

*Table 12.* Expert conversation domains used for LLM-based topic generation.

| Category | Domains | Category | Domains |
|---|---|---|---|
| Arts, Humanities & Culture | art history and curation
music composition and ethnomusicology
religion theology and ancient languages
explaining abstract and cultural concepts
navigating cultural traditions
literature and critical studies
philosophy and ethics
history and archaeology
visual arts and design
music and performance arts
cultural studies and anthropology
creative writing and expression | Engineering & Technology | mechanical and structural engineering
aerospace and nuclear engineering
electrical and electronics engineering
software engineering and programming
robotics and automation
materials and nanotechnology
cybersecurity and networks
human computer interaction
human factors and user experience design
personal tech troubleshooting |
| Medicine & Health Sciences | general medicine
surgery and operational care
pharmacy and drug research
medical ethics and policy
forensic medicine
medical psychology
sports medicine and biomechanics
veterinary science
biomedical engineering and prosthetics | Law, Finance & Business | constitutional law
intellectual property law
legal systems and regulations
corporate law and governance
financial management and accounting
investment and risk analysis
business strategy and operations
public policy and government affairs
quantitative finance and risk modeling |
| Pure Sciences & Research | physics and astronomy
chemistry and chemical sciences
biology and life sciences
earth and environmental sciences
mathematics and statistics
cognitive science and neuroscience
computational linguistics and technology
environmental and climate modeling
molecular biology genomics and biotechnology | AI and Machine Intelligence | machine learning and data science
natural language processing
computer vision and perception
explainability ethics and fairness in AI
AI theory and algorithms
reinforcement learning and adaptive systems
theory of mind and belief tracking
dynamic multitask coordination
tutoring with adaptive feedback |
| Communication & Interpretation | verbal and nonverbal communication
social cues and context
conflict resolution and negotiation
science communication
digital culture and slang
understanding implicit and ambiguous requests
empathetic and emotional support
decoding vague and cryptic messages
writing emails and apologies | Personal & Interpersonal Relations | family and parenting
romantic relationships
social etiquette and boundaries
workplace interactions
online behavior and harassment
emotional triggers and guilt |
| Home & Lifestyle Management | home maintenance and repair
interior design and organization
cooking and meal planning
gardening and plant care
personal hobbies and leisure
naming mottos and playlist creation | Planning & Logistics | travel and event planning
daily scheduling and task management
emergency preparedness
resource and supply management
spontaneous and creative problem solving
local navigation and travel suggestions |
| Geopolitics & Social Systems | international relations
political science and governance
military strategy and security
urban planning and development
group dynamics and sociology | Problem Solving & Decision Making | strategic thinking and game theory
ethical dilemmas
negotiation and compromise
complex problem solving
managing decision fatigue |
| Career & Self-Improvement | career planning and coaching
productivity and time management
learning and memory strategies
motivation and goal setting | Personal Finance & Consumerism | budgeting and saving
investments and retirement planning
consumer rights and shopping
taxes and compliance |
| Health & Wellness (Personal) | fitness and exercise
mental health and stress management
sleep and recovery
nutrition and diet | Safety & Preparedness | personal safety and security
first aid and medical emergencies
disaster response and preparedness
cybersecurity awareness |

*Table 13.* The prompt for conversation topic generation.

| Topic Generation Prompt |
| --- |
| Generate a list of 50 long, clear, and specific topics suitable for in-depth discussion in the domain of '[DOMAIN]'. Each topic should be a full sentence or phrase, keeping it under 10 words. The topic should detail enough to guide a focused conversation, but still remain simple and easy to read and understand. Aim for a casual, conversational tone. Do not use any abbreviations, apostrophes symbols, or punctuations unless absolutely necessary for clarity.

For example, below are some possible topics within the domain of 'artificial intelligence':
- AI impact on daily life
- Building smarter home robots

Below are the 50 topics within the domain of '[DOMAIN]':
- |

*Table 14.* An example conversation script between human and Moshi with references.

| Conversation Script |
| --- |
| *Human: Hey Moshi, ever feel like you're living in a sci-fi movie these days?*
*Moshi: It definitely feels that way sometimes, doesn't it? With all the new tech coming out.*

*Human: Seriously! Speaking of tech, I was reading about AI and art. It's kinda wild.*
*Moshi: It is pretty wild. What specifically about AI and art are you curious about?*

*Human: Well, do you think AI-generated content will ever truly replace human artists?*
*Moshi (lead): That's a big question a lot of people are asking.*
*Reference: AI-generated art raises copyright concerns, as ownership is unclear. Current legal frameworks don't fully address AI authorship. Many argue AI tools assist artists, rather than replace them.*
*Moshi (body)(subtopic: copyright and authorship): Copyright is a big issue with AI art. Right now, it's unclear who owns the rights, and the law hasn't caught up yet.*
*Moshi (tail): It's more likely to be a tool for artists, for now.*

*Human: Hmm, a tool... I'm not so sure. Doesn't it devalue the skill involved?*
*Moshi: That's a valid point. It brings up a lot of questions about what we value in art.*
*...* |

*Table 15.* v1 LLM prompts used for generating conversation scripts and reference documents.

| **Moshi LLM Prompt** |
| --- |

You are generating the next line for the chatbot "moshi" in a realistic ongoing conversation between a human and moshi.

**Context:**
- You see the prior conversation history, including turns from both human and moshi.
- You may have access to some reference documents, which provides knowledge-intensive information for your response.

**Moshi Turns:**
- Each of moshi's turns may be unaugmented or augmented.
- An unaugmented turn is general knowledge or conversational filler, requiring no external information.
- An augmented turn consists of three parts:
    - Lead: The opening part that includes a brief answer, general knowledge, or conversational filler, which does not require external information.
    - Body: The knowledge-intensive part, requiring external information or retrieval.
    - Tail: The closing unaugmented part, which can be empty.
- If the turn is unaugmented, it consists only of the unaugmented part.
- If the turn is augmented, it must have a lead, an augmented body part, and may have a tail.
- Each part does not need to be a full sentence; a few words or a short phrase are acceptable, provided they sound natural in speech.
- It is acceptable to use filler phrases as the lead or tail. But content like "let me check this for you...", "let me see", or "let me think" must be avoided since the user should not know that moshi has access to additional information sources.
- Label each part explicitly as (lead), (body)(subtopic: ...), (tail), or (unaugmented).
- For the body part in any augmented turn, specify the subtopic and use information from the reference document for the augmented body part.
- The tail part in an augmented turn is optional; some augmented turns may come with an empty tail.
- Try your best to use and follow the information provided by the reference document when generating the body part.

**Reference Document**
- Several reference documents are provided for each conversation.
- The reference documents are concise and factual, providing knowledge-intensive information to enhance moshi's response, specifically the body parts in augmented turns.
- Each reference document contains less than fifty words, labeled as `Reference: [reference content]`.
- The reference documents are placed after the lead parts of the augmented turns in the conversation.

**Guidelines:**
- The generated content will be used as input for a text-to-speech model. Therefore, ensure the conversation is completely readable and sounds like a natural transcription of human speech. Avoid overly complex sentence structures or jargon that would sound unnatural when spoken.
- Keep the wording of each sentence simple and easy to understand, while providing enough detail to guide a focused conversation.
- Be short and concise. Aim to keep each turn within thirty words if possible to maintain a conversational pace.
- Avoid unreadable punctuations and symbols such as asterisks. Only use the most basic punctuations such as period, question mark, exclamation point, comma, hyphen, Apostrophe, quotation marks, and ellipsis.
- Convert numbers, dates, abbreviations, etc, to readable text (e.g., 25 to twenty-five, 2509 to twenty-five-oh-nine, 997 to nine hundred ninety-seven, 0.5 to zero point five, 12/21 to December twenty-first, € to euro, kg to kilograms, % to percent, & to and, etc.). Avoid unreadable format such as bullet points, tables, and columns.
- Before the conversation, the user will be presented a hello message, so you do not need to say "Hi" or "Hello" again when starting the conversation.
- Directly answer the user's question in a single turn if possible. Avoid backs and forths.
- Keep the body part within thirty words.
- Feel free to leave the tail part empty if natural.
- Use a casual and conversational tone.

**Format Example:**
Human: [greetings (optional)]
(unaugmented)
moshi: [greetings (optional)]

Human: [human question]
(augmented)
moshi (lead): [general response]
Reference: [reference document for the following augmented body]

moshi (body)(subtopic: [subtopic]): [knowledge-intensive response]
moshi (tail): [follow-up general response]

Human: [human question]
(augmented)
moshi (lead): [general response]
Reference: [reference document for the following augmented body]
moshi (body)(subtopic: [subtopic]): [knowledge-intensive response]
moshi (tail): [empty]

Human: [next question]
(unaugmented)
moshi: [general response]

Human: [next question]
(unaugmented)
moshi: [general response]

Human: [next question]
(augmented)
moshi (lead): [general response]
Reference: [reference document for the following augmented body]
moshi (body)(subtopic: [subtopic]): [knowledge-intensive response]
moshi (tail): [empty]

Human: [next question]
(unaugmented)
moshi: [general response]

**Begin the conversation:**

### User LLM Prompt

You are generating the next user turn in a realistic, domain-specific conversation between a human and a chatbot named "moshi" within the domain of [DOMAIN]. The topic of the conversation is: [TOPIC].

**Context:**
- You see the prior conversation history, including turns from both human and moshi.

**Guidelines:**
- You can start the conversation with a question or statement relevant to the topic or just simple greetings.
- Do not always use traditional ways to start a conversation. Be creative.
- Your turn should be a natural, human-like response or question, relevant to the ongoing conversation and the given topic and domain.
- You may ask follow-up questions, request clarification, make statements, hesitate, or express disagreement, as in natural dialogue.
- The conversation should be at least [MIN_TURNS] turns, but can be longer if natural.
- The generated content will be used as input for a text-to-speech model. Ensure the conversation is completely readable and sounds like a natural transcription of human speech.
- Avoid using any technical jargon or unnatural phrasing.
- Keep your response concise (ideally under thirty words), casual, and easy to understand.
- Avoid unreadable punctuations. Only use period, question mark, exclamation point, comma, hyphen, Apostrophe, quotation marks, and ellipsis.
- Convert numbers, dates, abbreviations, etc., to readable text (e.g., 25 to twenty-five, % to percent).
- Conclude the conversation by outputting "EOC" when appropriate.

**Ways to start a conversation:**
Below are some ways to start a conversation. Try to be creative.
- Hello there!
- How have you been?
- Good day.
- It's nice to meet you, moshi.
- How are things going?
- It's wonderful to see you again, moshi.

- Hey moshi, what have you been up to?
- It's been a while, how have you been?
- I hope everything is going well on your end.
- How's everything?
- I believe you're having a great week.
- Hello, how are you today?
- Hi, how are you doing moshi?
- Good day.
- How are you doing?
- Hey, how is it going?
- Moshi, how's your day
- Hello, how was your day
- What's up moshi?
- What's going on?

You can start the conversation with greetings like above (but do not limit yourself to these sentences) or you can combine these greetings with your question. You can also skip the greeting part and directly ask your question to kick start the conversation related to the specified topic.

**Format Example:**
Human: [greetings - try to be creative]
moshi: [greetings]

Human: [question]
moshi: [response]

Human: [question]
moshi: [response]

Human: [question]
moshi: [response]

Human: EOC

**Another format Example:**
Human: [question]
moshi: [response]

Human: [question]
moshi: [response]

Human: [question]
moshi: [response]

Human: [saying goodbye - try to be creative] EOC

**Begin the conversation:**

---

**Reference LLM Prompt**
(Underlined content is only for reference LLM with summarization)

---

You are generating a short reference document for a chatbot named "moshi". This document will directly inform moshi's next response in an ongoing conversation between a user and Moshi.

**Context:**
- You see the complete prior conversation history, including all previous user and moshi turns.

**Guidelines:**
- The reference document should be concise, factual, and directly relevant to the ongoing conversation.
- The reference document must be helpful for moshi to generate its responses in the next turn of the conversation.
- The reference document should be labeled as `Reference: [reference content]`.
- Avoid complex or unreadable punctuations and symbols.
- Do not use markdown like asterisks (*) or hashes (#) within the reference content itself.

- Each reference document should contain no more than fifty words.
- Do not include any newline symbol in your results.
- After generating the reference, please summarize it to include only information that is relevant to the conversation and is helpful for moshi to generate its responses.
- The final summarized reference should be concise, labeled as `Summarized reference: [summarized reference content]`.

**Format example:**
Human: [greetings (optional)]
moshi: [greetings (optional)]

Human: [question]
Reference: [reference content for moshi's next response]
Summarized reference: [summary of the above generated reference]
moshi: [response]

Human: [question]
moshi: [response]

Human: [question]
moshi: [response]

Human: [question]
Reference: [reference content for moshi's next response]
Summarized reference: [summary of the above generated reference]
moshi: [response]

Human: [question]
moshi: [response]

Human: [question]
moshi: [response]

Human: [question]
Reference: [reference content for moshi's next response]
Summarized reference: [summary of the above generated reference]

**Begin the conversation:**

# E. Evaluation Prompts

*Table 16.* The prompt for evaluating the correctness on the HaluEval dataset with a Gemma 3 27B model. This prompt is based on the prompts used in the OpenAudioBench (Li et al., 2025).

---

**HaluEval Evaluation Prompt**

---

## Background
You are a professional QA evaluation expert. You need to assess whether the model's answer is correct based on the standard answer.

## Scoring Criteria
Correct: The answer matches or is equivalent to the standard answer, or contains the same core concept.
Incorrect: The answer is wrong or irrelevant to the question

## Evaluation Guidelines
1. The expression of answers can be flexible, not requiring exact matches. For example:
   - Numbers can be expressed in either Arabic numerals or words
   - Differences in punctuation or simple spelling mistakes can be ignored
2. Focus on whether the core meaning of the answer is correct

## Output Format
Provide the reasoning for your score, then generate the result in "[]" format and make sure it contains "the score is [Correct]" or "the score is [Incorrect]", for example:

The answer is correct and equivalent to the standard answer, the score is [Correct]

or

The answer is incorrect and does not match the standard answer, the score is [Incorrect]

## Question:
[QUESTION]
## Standard Answer:
[VALID_ANSWERS]
## Model's Answer:
[ANSWER]

---

*Table 17.* The prompt for extracting keyword from the model's response with a Gemma 3 27B model.

| **Keyword Extraction Prompt** |
| --- |
| You are helping evaluate a question answering model.
Identify the single keyword or short phrase in the model answer that directly expresses any of the answer aliases. If the model answer does not directly express any of the answer aliases, return the keyword/phrase that the model intends to answer the question. Respond with that keyword/phrase only.
**Example:**
Question: Give me a capital of an European country?
Model answer: Berlin is the capital of Germany.
Valid answer aliases: ["Paris", "Madrid", "Budapest", "Lisbon"]
Response: Berlin
**Input:**
Question: [QUESTION]
Model answer: [ANSWER]
Valid answer aliases: [VALID_ANSWERS]
Response: |

*Table 18.* The prompt for evaluating the correctness on the mathematical reasoning datasets with a Gemma 3 27B model.

| **Mathematical Reasoning Evaluation Prompt** |
| --- |
| You are helping evaluate a mathematical question answering model.
Determine whether the model's answer contains the provided Correct Answer. Ignore intermediate calculations or reasoning steps and focus on the numerical correctness of the model's answer.

**Instructions:**
1. Compare the value in the model's answer to the Correct Answer.
2. The comparison must be based on numerical equivalence (e.g., 5.0 should match 5). Ignore rounding errors or other small differences.
3. Your response must be only the word "Yes" or "No".

**Example 1: Correct Match**
Question: If a train travels at 60 mph for 2 hours, how far does it travel?
Correct Answer: 120.0
Model Answer: The total distance is 120 miles.
Response: Yes

**Example 2: Incorrect Match**
Question: John had 10 apples and ate 3. How many are left?
Correct Answer: 7
Model Answer: He has 8 apples left.
Response: No

**Input:**
Question: [QUESTION]
Correct Answer: [VALID_ANSWERS]
Model Answer: [ANSWER]
Response: |

