# OpenReview forum: "MoshiRAG: Asynchronous Knowledge Retrieval for Full-Duplex Speech Language Models"
_ICML.cc/2026/Conference — ICML 2026 regular_

### Official Review · Reviewer_8eDq · 2026-03-11

**Soundness:** 3
**Presentation:** 3
**Significance:** 2
**Originality:** 3
**Overall Recommendation:** 4
**Confidence:** 4

**Summary:**

The paper improves over Moshi, a full-duplex audio-language dialogue model by injecting external retrieved knowledge into the generation process. Specifically, the retrieved text is encoded by a fixed text encoder, and the representations are added to the input hidden states along with the original listening/speaking/text channels. Experimental results show that Moshi-RAG outperforms Moshi in factuality, while keeping decent End-to-end keyword delay (E2EKD), which is the total time from the end of the user’s query to the moment the keyword is mentioned in the model’s response.

**Compliance With Llm Reviewing Policy:**

Affirmed.

**Final Justification:**

please see Rebuttal Acknowledgement

**Key Questions For Authors:**

- Please answer to the weaknesses mentioned above.

**Limitations:**

yes

**Strengths And Weaknesses:**

Strengths:
- The paper proposes a novel method for incorporating retrieval agents into full-duplex chatbots, which enhance their factuality without sacrificing responsiveness.
- The paper provides comprehensive experiments and analysis showing the motivation of design choices, and the advantage of Moshi-RAG over vanilla Moshi in question answering and reasoning tasks.
- The paper is well-presented.

Weaknesses:
- The proposed text-encoder-based additive injection suffers limited generalization capability when retrieved documents are too long, as it compresses the retrieval result length with only 4 times. I think compressing to a fixed length could be more reasonable. Besides, the current gap between reference accuracy and response accuracy is large on reasoning-intensive tasks (Table 3), indicating that the method is less effective in enhancing Moshi's long-range reasoning capability.
- Moshi-RAG's interactive mode, which is learned from the crafted training data by the authors, has side-effects including increased latency to user's interruption (Table 2), and a ~2-second Pre-RAG coarse answer. This harms the soundness/necessity of using a full-duplex model design, as cascaded systems or time-division based duplex systems (e.g., Qwen2.5-omni) can reduce their KD by switching to a more ``aggressive'' answer style via SFT, thereby achieving similar E2EKD.

---

> ### Author Rebuttal · Authors · 2026-03-26
>
> We thank the reviewer for their recognition and their thoughtful comment. In our detailed responses, we provide technical evidence for our choice of compression methods and discuss the core value of MoshiRAG as a framework that shows competitive factualness as turn-based models while preserving the full-duplexity without introducing significant overhead.
>
> ## Strengths And Weaknesses
>
> > The proposed text-encoder-based additive injection suffers limited generalization capability when retrieved documents are too long, as it compresses the retrieval result length with only 4 times. I think compressing to a fixed length could be more reasonable. Besides, the current gap between reference accuracy and response accuracy is large on reasoning-intensive tasks (Table 3), indicating that the method is less effective in enhancing Moshi's long-range reasoning capability.
>
> **Response a:**
>
> We evaluated more aggressive compression ratios (results in Table 7), but found they consistently degraded overall QA performance. Consequently, we maintained a fixed compression ratio of 4. In the rare event of exceptionally long reference documents—which is mitigated by our generation prompts in Table 15 which advises the reference model to be consise—a pre-encoding summarization strategy in Table 3 can provide an effective fallback. For context, 2 seconds of Moshi’s audio stream corresponds to 25 tokens, which accommodates 100 reference text tokens at a compression ratio of 4, a sufficient window for most retrieved knowledge.
>
> The performance gap observed in mathematical reasoning tasks (Table 3) is primarily a consequence of these tasks being out-of-distribution for the current iteration of MoshiRAG. Since the model was trained exclusively on QA-style data, it has not yet been optimized to process dense reasoning chains or complex equations. While this gap could be narrowed by incorporating mathematical or multi-domain conversations into the training data, the results in Table 3 already demonstrate the architecture's strong potential to generalize to out-of-domain tasks without explicit task-specific training.
>
> > Moshi-RAG's interactive mode, which is learned from the crafted training data by the authors, has side-effects including increased latency to user's interruption (Table 2), and a ~2-second Pre-RAG coarse answer. This harms the soundness/necessity of using a full-duplex model design, as cascaded systems or time-division based duplex systems (e.g., Qwen2.5-omni) can reduce their KD by switching to a more ``aggressive'' answer style via SFT, thereby achieving similar E2EKD.
>
> **Response b:**
>
> While it is theoretically possible to improve the Keyword Delay (KD) of turn-based models through aggressive supervised fine-tuning (SFT), the fundamental advantage of full-duplex models like MoshiRAG lies in their high-resolution temporal modeling. By operating at the precision of a single speech codec frame, these models can react instantaneously to both verbal and non-verbal conversational events. Current benchmarks, such as interruption latency or E2EKD, only partially capture this capability. A full-duplex model can naturally achieve low E2EKD and low interruption latency but is not the only way to beat these benchmarks.
>
> As a result, the results in the paper are intended to demonstrate that MoshiRAG has competitive latency performance when compared against other duplex and non-duplex models without being specifically optimized for these metrics. The integration of RAG introduces measurable overhead (specifically the 2-second pre-RAG content and slight increases in interruption latency), but most importantly, the MoshiRAG framework preserves the core nature of full-duplexity. The system maintains its capacity for immediate responsiveness to user inputs, a structural advantage that existing benchmarks have yet to fully quantify but which remains a defining feature of the full-duplex architecture.
>
> Lastly, the delay metrics in Table 1 is computed with a single-turn testing scenario, but the real use case is multi-turn. Although when retrieval is triggered, the pre-RAG part does increase the delay of MoshiRAG, the retrieved information remains useful for later non-RAG turns in the conversation. The E2EKD for these later turns is expected to be less than 3.1 seconds as they do not have a pre-RAG part. In contrast, the delay of turn-based models is expected to be roughly the same for the first turn and later turns.

---

> > ### Author Rebuttal · Reviewer_8eDq · 2026-04-03
> >
> > The authors clarified on the solution for over-long sequences and the effects on latency, which helps the reviewers and readers better understand the work. I would maintain my overall recommendation of 4 based on my assessment of the paper as a whole.

---

### Official Review · Reviewer_8UkF · 2026-03-11

**Soundness:** 3
**Presentation:** 3
**Significance:** 3
**Originality:** 3
**Overall Recommendation:** 5
**Confidence:** 5

**Summary:**

This paper proposes a method to enable the full duplex model Moshi using retrieval augmented generation (RAG)

**Compliance With Llm Reviewing Policy:**

Affirmed.

**Key Questions For Authors:**

- What happens if another retrieve token is generated after the first generation? What if they are one word apart?
- Given that all data is synthetic, can you think of a non-synthetic dataset to augment? Is it too hard to collect?
- Why do you think TTS-generated speech is representative enough for training?

**Limitations:**

Only partially in the Conclusions.

**Strengths And Weaknesses:**

Strengths:
- Many real world scenarios require some sort of RAG augmented model using custom documents. This paper closes such an obvious gap for speech processing.
- The approach is sound, and experimental results are satisfactory
Weaknesses:
- Approach relies on finding the important word (The authors should also cite Shih paper here) to output the retrieve token. Experiments are then performed on question answering tasks. These tasks favor the approach. On the other hand there are many other conversational AI tasks beyond Q&A, such as customer support.
- Filler words for E2EKD seem ad hoc.

---

> ### Author Rebuttal · Authors · 2026-03-26
>
> We appreciate the reviewer’s positive assessment of MoshiRAG’s soundness and its importance in closing a critical gap in speech processing. We provide the detailed responses to address the reviewer’s concern about evaluation tasks, training dataset creation, and metrics below:
>
> ## Strengths And Weaknesses
>
> > Approach relies on finding the important word (The authors should also cite Shih paper here) to output the retrieve token. Experiments are then performed on question answering tasks. These tasks favor the approach. On the other hand there are many other conversational AI tasks beyond Q&A, such as customer support.
>
> **Response a:**
>
> We will include the suggested citation "Can speech LLMs think while listening?"
>
> While using QA as the primary evaluation focus due to the availability of widely-accepted established benchmarks, we agree that domains like customer support are vital. The current bottleneck is the lack of universal, standard benchmarks for the evaluation of voice assistants on general-purpose speech tasks, especially for the retrieval augmented scenario. We will closely follow the updates in the community and look for opportunities to evaluate MoshiRAG under more diverse scenarios and task categories.
>
> In the meantime, our upcoming live demo will serve as an initial validation of MoshiRAG’s versatility across diverse, user-defined topics.
>
> > Filler words for E2EKD seem ad hoc.
>
> **Response b:**
>
> To provide context for our metric selection, it is essential to distinguish between non-duplex and full-duplex speech models. Non-duplex models (such as Qwen Omni) lack the capability for concurrent input/output processing, often resulting in a perceptible "silent gap" between a user’s query and the model’s response. In contrast, full-duplex models continuously process and generate speech tokens without interruption, maintaining the natural cadence of human conversation.
>
> We utilize End-to-End Keyword Delay (E2EKD) specifically because it is a human-perceivable metric that accurately compares these two architectures. It measures the most critical factor for the user experience: the total time elapsed before the requested information is delivered.
>
> The use of pre-RAG "filler content" is a deliberate architectural compromise to bridge the retrieval latency. While non-duplex models remain silent during back-end processing, MoshiRAG leverages its full-duplex nature to keep the conversation active. We agree that current pre-RAG templates can produce user-perceptible repetitiveness; however, we believe that this is still preferable to unnatural silence. Future iterations will focus on increasing the diversity and naturalness of the pre-RAG content to further align with human temporal gap-filling behaviors.
>
> ## Key Questions For Authors
>
> > What happens if another retrieve token is generated after the first generation? What if they are one word apart?
>
> **Response c:**
>
> Repetitive `<ret>` token generation is rare because the training distribution lacks such patterns. However, the system architecture handles this by allowing the latest retrieval request to preempt any unfinished ongoing requests. This ensures the model always prioritizes the most recent and relevant query context.
>
> > Given that all data is synthetic, can you think of a non-synthetic dataset to augment? Is it too hard to collect?
>
> **Response d:**
>
> Collecting real-world data for this specific task is challenging because humans do not naturally retrieve and integrate external information as fast as machines do within the 2-second window. A possible approach is post-hoc labeling human conversations, adding reference information to knowledge-intensive turns in existing human conversations. However, this creates a mismatch between training and inference setups (generating response based on reference, or generating reference based on response). At this moment, we do not have an experiment to compare the trade-offs between these two strategies.
>
> > Why do you think TTS-generated speech is representative enough for training?
>
> **Response e:**
>
> While TTS-generated speech is not the optimal medium for building a full-duplex model, MoshiRAG benefits from being built upon the original Moshi architecture. During its pre-training phase, Moshi was exposed to extensive real-world human conversation data, allowing it to handle natural verbal cues and spontaneous interactions.
>
> We then utilized synthetic data specifically for the RAG post-training phase to teach the model how to effectively incorporate retrieved reference information. This hybrid approach demonstrates a critical design principle: leveraging massive, general-purpose human datasets for conversational fluency, while utilizing targeted, task-oriented synthetic data to optimize specific assistant behaviors.
>
> Upon the release of our live demo, users will be able to empirically validate MoshiRAG’s ability to maintain high-quality, real-time interactions in real-world, spontaneous environments.

---

> > ### Author Rebuttal · Reviewer_8UkF · 2026-04-04
> >
> > Thanks for the reply. I will keep my score as "Accept"

---

### Official Review · Reviewer_coto · 2026-03-12

**Soundness:** 4
**Presentation:** 4
**Significance:** 3
**Originality:** 3
**Overall Recommendation:** 5
**Confidence:** 4

**Summary:**

The paper introduces MoshiRAG, an extension to the Moshi full-duplex speech language model that incorporates retrieval-augmented generation (RAG) capabilities. To circumvent the strict real-time constraints of full--uplex interaction, the authors propose an asynchronous framework that fetches external knowledge while the model generates conversational fillers or coarse answers, referred to as pre-RAG content. By exploiting the natural "keyword delay" time between the onset of the model's speech and the delivery of the core factual answer, MoshiRAG integrates retrieved text before the main response is spoken. The system demonstrates significantly improved factuality on speech QA benchmarks and generalizes well to mathematical reasoning tasks, all while maintaining the low latency and interactivity typical of full-duplex models.

**Compliance With Llm Reviewing Policy:**

Affirmed.

**Final Justification:**

Given the technical merit and the resolution of the most pressing concerns, I'm keeping my recommendation of Accept (5).

**Key Questions For Authors:**

1. How does the system gracefully handle edge cases in real-world deployments where the retrieval delay unexpectedly exceeds the duration of the pre-RAG content?

2. Given the model's high sensitivity to ASR Word Error Rates, have you considered leveraging the model's internal acoustic representations directly for retrieval instead of relying on a separate ASR model?

3. You cite information loss during knowledge integration as a current limitation. Are there specific architectural changes or attention mechanisms planned for future iterations to mitigate this beyond the current strategy?

**Limitations:**

The authors adequately discuss the limitations of their work. They explicitly note the system's reliance on a separate ASR model and analyze its high sensitivity to transcription errors. They also recognize that the current retrieval trigger relies entirely on synthetic training data distributions

**Strengths And Weaknesses:**

Strengths

Soundness: The timing constraints are rigorously analyzed, and the training data generation realistically simulates retrieval delays to ensure robustness. Experimental results clearly validate both the factuality improvements and the preservation of interactivity, as evidenced by lower takeover rates during user interruptions.

Presentation: The authors do an excellent job defining critical and potentially confusing latency metrics early in the paper, such as Time-to-first-audio-token (TTFAT), Keyword delay, and End-to-end keyword delay (E2EKD). This establishes a clear foundation for understanding the system's timing constraints. The manuscript features highly informative diagrams that clarify complex concepts. For example, Figure 3 and Figure 4 effectively illustrate the asynchronous interaction between the front-end token streams and the back-end retrieval system. The Related Work section properly positions the research within the current landscape of speech-to-speech models, clearly distinguishing the unique challenges of real-time full-duplex RAG from existing turn-based approaches and concurrent works like Stream RAG. The appendices are exceptionally thorough, providing the exact prompts used for data generation and evaluation, along with comprehensive data statistics and specific training hyperparameters. This level of detail significantly aids in reproducibility.

Significance: The approach addresses a critical limitation regarding the factuality and hallucination rates of native audio models. The modular design allows for plug-and-play retrieval backends (such as search engines or different LLMs) without retraining the base model, offering high flexibility.

Originality: This work represents the first attempt to integrate RAG into a full-duplex speech language model under strict real-time constraints. The strategy of using the "keyword delay" to asynchronously mask retrieval latency is a highly practical and clever innovation.

Weaknesses

Soundness: The system is highly sensitive to automatic speech recognition (ASR) errors, which significantly degrades the accuracy of the retrieved references and final responses. A persistent gap between retrieved reference accuracy and final response accuracy indicates that there is noticeable information loss during the knowledge retrieval phase.

Presentation: While generally well-structured, the paper relies heavily on synthetic multi-turn conversation scripts, and the transition of this synthetic training to real-world, highly spontaneous queries could be more thoroughly detailed.

Significance: The system heavily relies on generating "pre-RAG" content, such as conversational fillers, to mask the retrieval latency. While effective for timing, repeatedly relying on this specific conversational template could lead to a formulaic and repetitive user experience in real-world deployments, potentially detracting from naturalness. Although the authors position this in the context of advanced speech LMs, the system still fundamentally relies on a text-based streaming ASR model to transcribe user queries for the retrieval backend. The paper acknowledges that the model is highly sensitive to ASR errors, which can severely degrade the accuracy of the final response. This reliance on an explicit text transcription step limits the significance of the end-to-end audio. The paper notes a persistent gap between the accuracy of the retrieved reference documents and the accuracy of Moshi's final spoken response. This indicates that the current additive information injection strategy struggles to fully capitalize on the provided external knowledge, limiting the overall impact of the RAG integration.

Originality: The system design is primarily an integration of existing, off-the-shelf components. It combines the existing Moshi model, a pre-trained streaming ASR model, an existing sequence compression network, and existing text-based LLMs or search engines for the backend. The architectural modifications to Moshi itself are minimal, consisting mostly of introducing a single retrieval token and a reference text encoder. The core insight of using a "keyword delay" to asynchronously fetch information while generating filler text is a well-established technique in text-based conversational agents. While applying it to the strict timing constraints of a full-duplex speech model is a novel engineering feat, the underlying conceptual mechanism is not entirely new. The authors properly cite concurrent work, such as Stream RAG, which similarly exploits temporal gaps in spoken conversations to perform information retrieval. While MoshiRAG distinguishes itself by operating in a strict full-duplex setting, the existence of parallel research exploring the same delay mechanism slightly diminishes the originality of the core idea.

---

> ### Author Rebuttal · Authors · 2026-03-26
>
> We thank the reviewer for their detailed and encouraging feedback. Detailed responses to the questions of the reviewer are provided below:
>
> ## Strengths And Weaknesses
>
> > Soundness: The system is highly sensitive...
>
> **Response a:**
>
> The MoshiRAG retrieval pipeline consists of three stages: (1) ASR (User speech to text), (2) Information Retrieval, and (3) Knowledge Integration into the final audio response. While this modularity facilitates "plug-and-play" flexibility for various retrieval backends, we acknowledge that each interface introduces a potential for information loss.
>
> As noted, the performance gap between retrieved reference accuracy and final response accuracy (Stage 3), compounded by streaming ASR errors (Stage 1), represents a clear area for optimization. This compound error currently acts as a performance ceiling for MoshiRAG. Enhancing the system involves two primary trajectories:
>
> - Module-Level Optimization: Integrating lower-latency, higher-accuracy streaming ASR models or developing more sophisticated information injection strategies that surpass the current ARC-Encoder’s capabilities.
> - Joint Optimization: Developing retrieval methods inherently robust to ASR noise, or conditioning information compression on the immediate conversational context to prioritize the most relevant data.
>
> While these specific optimizations are outside the scope of the current manuscript, they represent vital next steps in making full-duplex models more resilient to conversational variance and more efficient in integrating external knowledge.
>
> ## Key Questions For Authors
>
> > How does the system gracefully handle edge cases in real-world deployments where the retrieval delay unexpectedly exceeds the duration of the pre-RAG content?
>
> **Response b:**
>
> The additive injection mechanism continues to process the embedding sequence until completion, regardless of whether the pre-RAG content is completed or not. During training, to ensure system stability, we applied a 0.2 dropout rate to the reference information. This forces the model to maintain a coherent conversation even if retrieval information does not arrive in place in time. In such cases, the model will keep generating its response without reference information to maintain conversational continuity despite that the correctness of the content generated in this situation will be suboptimal – which is the reason why we aim to avoid this situation in any case.
>
> > Given the model's high sensitivity to ASR Word Error Rates, have you considered leveraging the model's internal acoustic representations directly for retrieval instead of relying on a separate ASR model?
>
> **Response c:**
>
> We did consider augmenting Moshi with a parallel user-text stream to replace the external ASR model, which would have provided synchronized audio and text streams for both the user and the assistant. This approach was intended to eliminate discrepancies between Moshi’s audio perception and the ASR transcription—such as cases where Moshi hears a phonetic variation (e.g., "for all intensive purposes") while the ASR correctly transcribes the intended phrase ("for all intents and purposes").
>
> However, we opted against this because incorporating a native text stream significantly increases training difficulty. Also, even without mismatched transcriptions, the system is still vulnerable to ASR errors when Moshi fails to accurately transcribe the user’s speech. Given these factors, utilizing a mature, high-performance streaming ASR model is a more robust and practical solution.
>
> Furthermore, we chose not to use internal acoustic representations for retrieval to maintain "plug-and-play" modularity. Since most existing retrieval back ends and search engines are built for text interfaces, a text-based bridge ensures maximum compatibility and flexibility.
>
> > You cite information loss during knowledge integration as a current limitation. Are there specific architectural changes or attention mechanisms planned for future iterations to mitigate this beyond the current strategy?
>
> **Response d:**
>
> Preliminary studies (Appendix Table 6) indicate that insertive injection can outperform additive injection. We will investigate the feasibility of high-fidelity insertive injection without sequence compression to minimize information loss while balancing memory overhead.
>
> Cross-attention mechanisms were tested early in development but consistently underperformed relative to the insertive and additive injection strategies in the paper.

---

> > ### Author Rebuttal · Reviewer_coto · 2026-04-02
> >
> > Handling Retrieval Delay Edge Cases: Your explanation regarding the 0.2 dropout rate during training is a practical and sound approach. It makes sense that forcing the model to occasionally operate without reference information ensures it can gracefully fall back to maintaining conversational continuity, even if the factuality is suboptimal in those specific edge cases.
> >
> > ASR vs. Acoustic Representations: You provided a strong justification for relying on an external streaming ASR rather than native acoustic representations. I agree that maintaining a text-based bridge is the most practical choice right now, as it preserves the highly valuable "plug-and-play" modularity with existing text-based search engines and LLMs.
> >
> > Information Loss Mitigation: Pointing to the preliminary studies on insertive injection and clarifying that cross-attention consistently underperformed provides excellent context regarding your architectural explorations and future optimization trajectories.
> >
> > Overall, the rebuttal reinforces my positive assessment of the paper. You have demonstrated a good understanding of the system's current limitations, specifically the performance ceiling created by compound ASR and injection errors, while presenting an original and effective framework.
> >
> > I will be maintaining my original score of 5: Accept.

---

### Decision · Program_Chairs · 2026-04-30

**Decision:**

Accept (regular)

**Comment:**

MoshiRAG incorporates retrieval-augmented generation (RAG) into the Moshi full-duplex speech model. The architectural framework includes a <ret> token for query identification, an ARC-Encoder designed to compress documents into prefixes, and a streaming ASR backend. Utilizing synthetic multi-turn post-training, the model develops the capability to determine appropriate retrieval instances and articulate retrieved summaries. It sustains low latency while attaining factuality benchmarks comparable to leading non-duplex speech language models. Evaluations conducted on question-answering datasets, mathematical reasoning, and interactivity data demonstrate that MoshiRAG achieves factuality while maintaining the low-latency, interruptible nature of full-duplex dialogue.


There is a consensus among reviewers that MoshiRAG effectively addresses a significant and well-defined challenge: the integration of factual grounding into full-duplex speech dialogue systems without compromising their characteristic low-latency interactivity. The fundamental approach—leveraging keyword delay to asynchronously retrieve knowledge—represents an innovative and practically viable design. This research is technically rigorous, clearly articulated, and reproducible due to the inclusion of detailed appendices, while demonstrating substantial empirical improvements.